# HA-VLN 2.0: An Open Benchmark and Leaderboard for Human-Aware Navigation in Discrete and Continuous Environments with Dynamic Multi-Human Interactions

## Abstract

Vision-and-Language Navigation (VLN) has been studied mainly in either *discrete* or *continuous* settings, with little attention to dynamic, crowded environments. We present HA-VLN 2.0, a unified benchmark introducing explicit social-awareness constraints. Our contributions are: (*i*) a standardized task and metrics capturing both goal accuracy and personal-space adherence; (*ii*) HAPS 2.0 dataset and simulators modeling multi-human interactions, outdoor contexts, and finer language–motion alignment; (*iii*) benchmarks on *16,844* socially grounded instructions, revealing sharp performance drops of leading agents under human dynamics and partial observability; and (*iv*) real-world robot experiments validating sim-to-real transfer, with an open leaderboard enabling transparent comparison. Results show that explicit social modeling improves navigation robustness and reduces collisions, underscoring the necessity of human-centric approaches. By releasing datasets, simulators, baselines, and protocols, HA-VLN 2.0 provides a strong foundation for safe, socially responsible navigation research.[1]

## 1 Introduction

Vision-and-Language Navigation (VLN) Anderson et al. (2018); Zhang et al. (2024b) challenges embodied agents to interpret natural-language instructions and reach specified goals in photorealistic simulators or real-world environments Gu et al. (2022); Wang et al. (2022). Although recent advances have delivered strong performance in controlled benchmarks, existing methods are typically confined to either *discrete* (DE) or *continuous* (CE) settings, neglecting the complexities of crowded, human-populated spaces, where agents must contend with unpredictable human behaviors, reason under partial observability, and ensure socially compliant navigation Anderson et al. (2021); Kadian et al. (2020); Yu et al. (2024). Bridging these gaps is essential for moving VLN from simulation prototypes toward robust real-world deployment Wu et al. (2024); Gao et al. (2024).

**Motivation and Open Challenges.** Despite recent progress, VLN research still faces three fundamental limitations that restrict its real-world applicability. First, *social awareness* remains under-explored: human participants in the scene are commonly overlooked or reduced to inert obstacles, preventing the agent from respecting personal space or reacting to bystanders' activities (see Figure 1). Second, *finer-grained instructions* are not well captured in existing corpora Paduraru et al. (2021); Kong et al. (2024). Commands such as "Turn to your left, and go past the chair" rarely reflect real-world contexts like "Turn to your left, where you will see someone taking a brief pause ... on the chair" in Figure 1. Third, *static-environment assumptions* neglects real-time re-planning when people traverse corridors or gather spontaneously. In practice, social navigation demands partial observability and dynamic route adjustment. Addressing these issues requires a benchmark that unifies DE and CE with explicit regime disclosure, supports socially grounded finer-grained instructions, and incorporates human-centric metrics for navigation in dynamic multi-human environments.

**Toward Human-Aware VLN.** Early progress, notably HA-VLN 1.0 framework Li et al. (2024) introduced dynamic humans into VLN, yet several shortcomings limited its realism and reproducibility (Appendix Table A1). Motion data in HAPS 1.0 Li et al. (2024) suffered from *alignment errors*

---

[1] **Project page:** `https://ha-vln-webpage.vercel.app/`

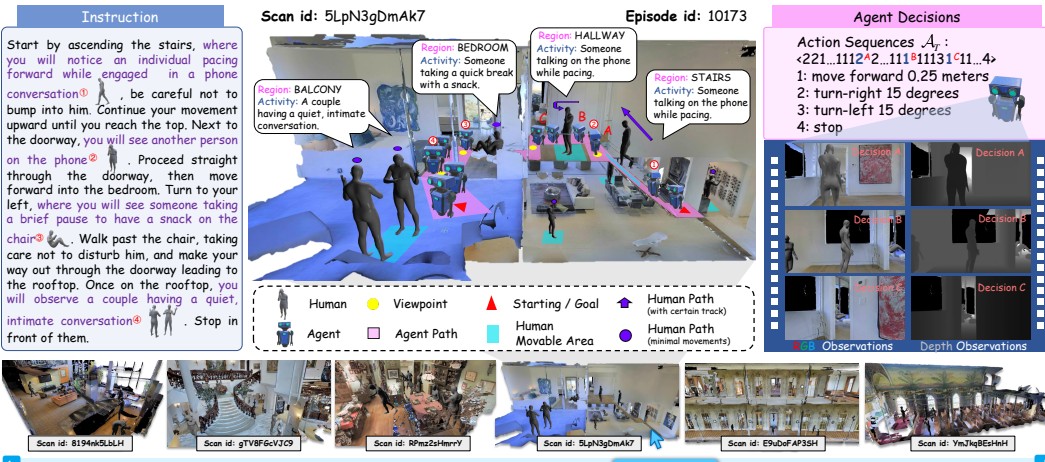

Figure 1: **HA-VLN 2.0 Navigation Scenario.** HA-VLN 2.0 adds four key challenges: (i) unified discrete/-continuous navigation with denser crowds, richer activities, and mixed indoor–outdoor scenes; (ii) stricter social-distance and collision constraints under partial observability; (iii) instructions explicitly grounded in human activities and spatial cues, improving language–vision alignment; and (iv) robust real-time planning amid occlusion and multi-human dynamics. Example: key positions (e.g., ①, ②) align with *instructional cues* referring to specific human behaviors. When the agent encounters a bystander on the phone (②, Decision A), it intelligently turns right to avert a potential collision. On the right, RGB and Depth observations illustrate the agent's panoramic view preceding decisions A, B, and C, capturing its dynamic responses to nearby humans.

*and limited diversity*, restricting coverage of everyday activities. The benchmark also exhibited a *discrete-navigation bias*, with its simulator largely confined to viewpoint hops Krantz et al. (2020) rather than physics-consistent low-level control Krantz et al. (2021). Multi-human interactions were *underdeveloped*, typically modeling only a single individual in simplified scenarios. Finally, instruction generation remained *coarse and object-centric*, omitting temporally varying activities and offering little control over granularity. These limitations call for a benchmark that standardizes regime disclosure, expands motion fidelity and diversity, incorporates multi-human interactions, and supports *finer-grained socially grounded instructions* across both discrete and continuous settings.

**Our Contributions.** In response, we introduce **HA-VLN 2.0**, a unified benchmark coupling discrete (DE) and continuous (CE) navigation paradigms with explicit social-awareness constraints. It comprises the *HAPS 2.0* dataset, featuring 486 SMPL-based motion sequences across 26 regions and 90 scenes, rigorously annotated via multi-view verification ( 430 annotation hours). HA-VLN 2.0 includes established simulators (HA-VLN-DE, HA-VLN-CE) with outdoor environments, real-time rendering, and precise collision management for up to 910 interacting humans. A unified API enables seamless comparisons across modes (Fig. 2; Sec. 3). Additionally, we expand R2R-CE Krantz et al. (2020) with 16,844 socially grounded instructions and introduce two baseline agents, HA-VLN-VL with Transformer-based grounding and HA-VLN-CMA with cross-modal attention for replanning, both validated under comprehensive human-centric metrics (Sec. 4). Finally, we demonstrate successful sim-to-real transfer and provide a public evaluation leaderboard (Sec. 5.2).

Specifically, HA-VLN 2.0 offers four key advancements:

1. **Cross-paradigm task standardization & Metrics.** We unify DE and CE navigation under social-awareness constraints, ensuring consistent goals and human-centric evaluations (Sec. 2).

2. **HAPS 2.0 & Dual simulators (large-scale build).** We release HAPS 2.0 (486 SMPL sequences) and two established simulators (HA-VLN-DE, HA-VLN-CE) that incorporate multi-view human annotation ($\sim$ 430 human-hours), outdoor scenes, dual-thread rendering, and rigorous collision checks for up to 910 active individuals with interactions (Fig. 2; Sec. 3).

3. **Comprehensive benchmarking with finer-grained instructions.** We augment R2R-CE with 16,844 socially-grounded instructions and benchmark multiple agents under unified metrics, unveiling challenges arising from multi-human dynamics and partial observability. (Sec. 4).

4. **Real-robot validation and public leaderboard.** We robustly demonstrate sim-to-real transfer using a physical robot successfully navigating crowded indoor areas, and provide a public leaderboard for comprehensive discrete–continuous evaluations in multi-human scenarios (Sec. 5.2).

## 2 THE UNIFIED HUMAN-AWARE VLN TASK

**Motivation and Overview.** HA-VLN 1.0 Li et al. (2024) introduced dynamic humans into VLN, but its discrete-environment (DE) focus limited ecological validity and hindered systematic study of continuous control and realistic multi-human interactions. To address this, we formalize *HA-VLN 2.0*, a unified benchmark that integrates DE and CE under explicit human-centric constraints. Under this setting, agents must parse instructions that reference ongoing human activities (e.g., *"Go upstairs where someone is pacing on the phone"*), anticipate plausible human trajectories, maintain socially compliant distances, and adapt plans online in densely populated, photorealistic 3D scenes (Fig. 1). We next make this specification precise by unifying state and action across regimes.

**Unified State and Action Spaces.** HA-VLN 2.0 defines a shared state and action interface bridging DE and CE. At each timestep $t$, the agent state is

$$s_t = \langle \mathbf{p}_t, o_t, \Theta_t^{\text{FOV}} \rangle, \tag{1}$$

where $\mathbf{p}_t$ is the agent's 3D position, $o_t$ its orientation, and $\Theta_t^{\text{FOV}}$ its egocentric visual observation. In DE, agents hop among predefined viewpoints with RGB observations; in CE, they perceive RGB+D within a $90°$ field of view and execute fine-grained increments (e.g., $0.25\,\text{m}$ forward, $15°$ rotation). Crucially, DE and CE now share a unified action space,

$$\mathcal{A} = \{a_{\text{forward}}, a_{\text{left}}, a_{\text{right}}, a_{\text{up}}, a_{\text{down}}, a_{\text{stop}}\}, \tag{2}$$

enabling direct and fair comparison across paradigms (Fig. 2).

**Human-Aware Enhanced Constraints.** HA-VLN 2.0 extends far beyond HA-VLN 1.0's sparse, static settings by introducing unified constraints that substantially increase realism and complexity in both DE and CE: (*i*) *Dynamic Human Models*: continuous trajectories from the HAPS 2.0 dataset capturing diverse behaviors and dense crowds; (*ii*) *Personal-Space Enforcement*: standardized proximity thresholds (3 m in DE; overlapping radii in CE) to ensure equitable cross-paradigm evaluation; (*iii*) *Human-Focused Instructions*: natural-language directives grounded in dynamic human behaviors, requiring precise alignment between text and visual context. All annotations are curated through a validated multi-stage pipeline (Sec. 3), ensuring both realism and reproducibility.

**Unified Dynamics and Partial Observability.** HA-VLN 2.0 formalizes a unified partially observable Markov decision process (POMDP) spanning both DE and CE settings, whereas HA-VLN 1.0 considered partial observability only in DE. Successor states $s_{t+1}$ depend jointly on agent actions and stochastic human dynamics (e.g., sudden path blockage or unexpected entry). Agents must therefore infer latent human intentions and strategically balance *exploration* (discovering alternate routes) with *exploitation* (committing to viable trajectories), reflecting the fundamental trade-offs inherent in navigation through dynamic, human-populated environments.

**Key Challenges of DE–CE Synergies.** Unifying DE and CE exposes three challenges for socially intelligent navigation: (*i*) *Socially Compliant Navigation*: collision-free movement that adapts to evolving personal-space boundaries; (*ii*) *Human-Aligned Instruction Grounding*: accurate interpretation of natural-language instructions amid dynamic human activities; (*iii*) *Adaptive Path Replanning*: trajectory adjustment in response to human interactions that modify accessibility. DE supports rapid prototyping and large-scale evaluation, while CE offers motion fidelity indispensable for bridging simulation and real-world deployment. Together, these synergies establish HA-VLN 2.0 as the first benchmark uniting efficient simulation with realistic human-populated environments, motivating a unified simulator and corresponding agent framework introduced next.

## 3 HA-VLN SIMULATOR

To support the unified HA-VLN task, we build a simulator that embeds dynamically moving humans in both *discrete* and *continuous* 3D environments. Unlike Li et al. (2024), which treated humans as static obstacles, our simulator models high fidelity motions, interactions among multiple humans, and socially grounded dynamics such as spontaneous movements, group activities, and personal space constraints. Using the upgraded HAPS 2.0 dataset, it improves motion diversity, spatial alignment, and realism over HAPS 1.0 (Table A2) and provides 486 curated sequences across indoor and outdoor scenes. The system includes two modules, HA-VLN-CE (continuous) and HA-VLN-DE (discrete), with a unified API (Sec. 3) for human state queries, dynamic scene updates, and collision

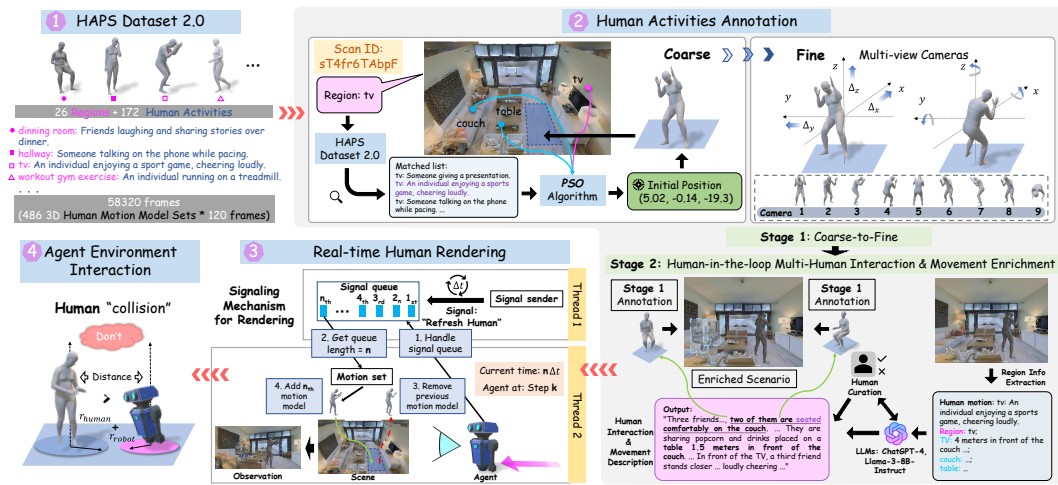

**Figure 2: HA-VLN Simulator.** Unlike HA3D, which modeled sparse and static human activities in discrete settings, HA-VLN incorporates *rich and dynamic* human behaviors using HAPS 2.0 (172 activities, 486 models, 58k frames). Annotation involves two stages: *(i) coarse-to-fine* optimization via PSO and multi-view camera setups, and *(ii) human-in-the-loop* refinement for realistic crowd dynamics. Real-time rendering updates motions through a signaling mechanism, facilitating collision detection and dynamic agent–environment interactions. These improvements bridge discrete evaluation (DE) and realistic continuous navigation (CE), establishing a robust foundation for benchmarks in socially intelligent navigation.

checks. Fig. 2 places these components in the agent's action and observation loop, forming the basis for the annotation, rendering, and interaction mechanisms that follow.

**HAPS 2.0 Dataset.** Human motion naturally adapts to and interacts with surrounding environments. The Human Activity and Pose Simulation (HAPS) Dataset 2.0 extends HAPS 1.0 Li et al. (2024) with two major advances: *(i) refined and diversified human motions* and *(ii) region-aware activity descriptions* (details in Sec. B.1). HAPS 2.0 defines 26 regions across 90 architectural scenes and contributes 486 validated activity descriptions covering indoor and outdoor contexts. These descriptions, verified by human surveys and quality control using ChatGPT-4 Brown et al. (2020), explicitly ground actions in regions (e.g., "workout gym exercise: an individual running on a treadmill"). The Motion Diffusion Model (MDM) Guy et al. (2022), built on the SMPL framework, converts these descriptions into 486 3D human motion models $\mathbf{H}$, yielding 120-frame sequences $\mathcal{H} = \langle h_1, h_2, \ldots, h_{120} \rangle$ that capture fine-grained motion and shape information[2]. Fig. A2 illustrates representative contexts, while Fig. A3 shows sample motions (e.g., climbing stairs, running).

**Human Activity Annotation: Coarse-Level.** To integrate HAPS 2.0 into our simulator, we adopt a coarse-to-fine strategy. At the coarse level, each region $\mathbf{R}$ is defined by a label $r$, boundary coordinates $\mathbf{B}_{lo} = (x_{lo}, y_{lo}, z_{lo})$ and $\mathbf{B}_{hi} = (x_{hi}, y_{hi}, z_{hi})$, and an object set $\mathbf{O} = \{j_1, j_2, \ldots, j_n\}$ with positions $\mathbf{p}^{j_i}$. We filter $\mathbf{H}$ to retain motions consistent with $r$, forming $\mathbf{H}'$. Each motion $h_i \in \mathbf{H}'$ is paired with an object $j_i \in \mathbf{O}$ via semantic similarity, producing $(h_i, j_i)$ pairs. Particle Swarm Optimization (PSO) Kennedy & Eberhart (1995) (Alg. A1) then determines the optimal placement $\mathbf{p}^{h_i}_{opt}$ around $j_i$, bounded by $\mathbf{R}$ and penalized if violating constraints such as maintaining a minimum distance $\epsilon = 1m$ from other objects or leaving the region (details in Appx. B.2). This yields natural placements that reflect realistic social behaviors and spatial relations.

**Human Activity Annotation: Fine-Level.** Building on coarse placements, fine-level annotation refinement leverages multi-camera observations, ensuring precise alignment of motions with scene geometry. Inspired by 3D skeleton capture systems Ji et al. (2018); Petrovich et al. (2021), we deploy nine RGB cameras around each human model (Fig. 2; see also Fig. A1). Each camera is located at $\mathbf{p}_{cam}$, shifted by $(\Delta_x, \Delta_y, \Delta_z)$ from the human position $\mathbf{p}_h$, with rotation angles $\theta_{lr}$ and $\theta_{ud}$. Horizontal shifts are set as $\Delta_x, \Delta_y = \epsilon$ and the vertical shift as $\Delta_z$. For camera $i$ ($i = 1, \ldots, 8$), $\theta^i_{ud}$ is defined as: $\tan \theta^i_{ud} = \begin{cases} 0 & : i \text{ is odd} \\ \frac{\Delta_z}{\sqrt{2}\epsilon} & : i \text{ is even} \end{cases}$ and the left-right angle $\theta^i_{lr} = \frac{\pi i}{8}$, while the overhead camera ($i = 9$) has $\theta^9_{lr} = 0$ and $\theta^9_{ud} = \frac{\pi}{2}$. This multi-view setup provides dense RGB coverage,

---

[2]$\mathbf{H} \in \mathbb{R}^{486 \times 120 \times (10+72+6890\times3)}$: 486 models × 120 frames with shape, pose, and mesh vertices.

enabling fine adjustments to resolve inconsistencies like mesh–object clipping. This stage took over 430 hours of annotation, yielding 529 models across 374 regions in 90 scans.

**Human Activity Annotation: Multi-Human Enrichment.** In Stage 2 (Fig. 2), we enrich scene diversity and interactions through a human-in-the-loop approach Ding et al. (2024), adding new characters and complex motion paths into regions $\mathbf{R}$ with existing activities $h_i$ at positions $\mathbf{p}^{h_i}$. Regional context, including objects $\mathbf{O}$ within 6 meters of $h_i$ and their positions, is provided to LLMs to generate diverse multi-human scenarios, which are refined in four rounds of manual review for scene consistency. Based on curated descriptions, new motions are placed relative to objects and annotated using the multi-camera method from Stage 1, enabling complex actions such as walking downstairs (details in Appx. B.5). After two annotation stages, the dataset comprises 910 human models across 428 regions in 90 scans (Fig. 3(a)(b)), including 111 outdoor humans, 72 two-person interactions, 59 three-person groups, and 15 four-person groups. Among these, 268 involve complex motions such as climbing stairs, substantially enriching the dataset with realistic behaviors. Detailed statistics are provided in Appx. B.8. This two-stage system enables precise modeling of social interaction spaces and personal boundaries, supporting agents in learning socially appropriate navigation strategies.

**Real-Time Rendering & Agent Interaction.** Beyond static annotation, our simulator continuously renders human motions in real time. A dual-thread producer–consumer architecture (Alg. A2) manages frame updates: Thread 1 enqueues refresh signals, while Thread 2 synchronizes with the agent's action cycle to process them. Each motion spans up to 120 frames; upon receiving a signal, Thread 2 discards outdated meshes and loads new ones, keeping retrieval delays below 50 ms. Fig. A2 illustrates how multiple humans are simultaneously maintained in a shared environment.

To close the loop, agents perceive these dynamics through a navigation mesh (navmesh) Savva et al. (2019). Collisions are flagged when bounding volumes overlap, i.e., when inter-object distances fall below the sum of their radii, triggering an automatic revert. This integration ensures agents not only experience dynamic and socially realistic environments but also learn to respect personal space and navigate effectively in dense human crowds.

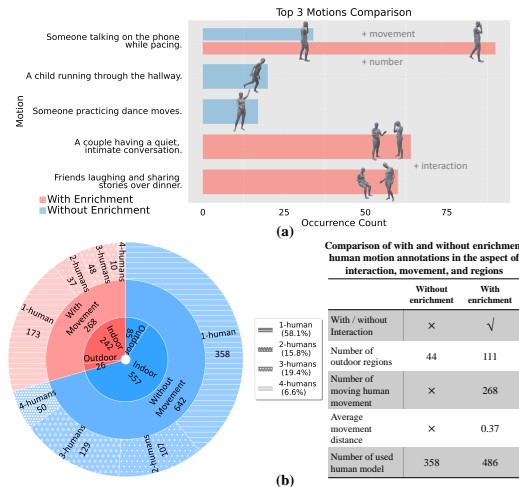

Figure 3: **Motion Analysis.** **(a)** Top three motions from Stage 1 (*without* enrichment) and Stage 2 (*with* enrichment). **(b)** Overall activity statistics, comparing interaction types, movement distances, and the number of models. Enrichment expands both the variety and dynamic range of human activities.

**Discrete vs. Continuous Settings.** **HA-VLN-CE** (Continuous) allows agents to move in real-valued increments (e.g., $0.25\,\mathrm{m}$ forward, $15°$ turns), supporting fine-grained collision avoidance and adaptive social behavior. As shown in Fig. A4, each scene can host up to 10 humans, with simulation speeds of 30–60 FPS on standard GPUs. **HA-VLN-DE** (Discrete) extends HA3D Li et al. (2024) by incorporating HAPS 2.0 data across indoor and outdoor environments. Agents hop among panoramic viewpoints while humans move continuously, preserving core social-navigation challenges. To align with continuous motions, we map positions to discrete nodes Li et al. (2024), apply small offsets for refinement, and integrate 627 annotated humans across 90 buildings.

**Unified API.** We provide a unified API supporting both modes with three core functions: (i) *Human State Queries* for retrieving bounding volumes, motion frames, and semantic annotations of nearby humans; (ii) *Dynamic Scene Updates* to notify agents of newly moved humans or environmental changes; and (iii) *Collision Checks* to evaluate whether a proposed move (e.g., forward step or viewpoint hop) would intersect with a human. By integrating HAPS 2.0, coarse-to-fine annotation, real-time multi-human rendering, and a single API across discrete and continuous settings, the HA-VLN Simulator establishes a comprehensive testbed for socially aware navigation. Figs. A2, A3, and A4 showcase the simulator's ability to capture diverse human behaviors, while Tables A1 and A2 highlight its advantages over prior simulators and the improvements of HAPS 2.0 relative to HAPS 1.0. Appendix B.7 provides details on environment scales, latency, and usage examples.

## 4 HA-VLN AGENTS

To ground the unified HA-VLN task in our HA-VLN simulator, we introduce the Human-Aware Room-to-Room (HA-R2R) dataset and two baseline agents, HA-VLN-VL and HA-VLN-CMA. These agents are designed as reference implementations rather than final solutions, offering a starting point for developing more advanced models. They emphasize essential social capabilities including maintaining personal space, avoiding collisions, and adapting to bystanders, under the dynamic conditions of HA-VLN 2.0. As shown in Figs. A8 and 5, human behaviors add substantial complexity, motivating the dataset design and agent baselines described in the following paragraphs.

**HA-R2R Dataset.** The Room-to-Room in Continuous Environment (R2R-CE) dataset Krantz et al. (2020) supports continuous navigation but lacks explicit modeling of human interactions. We therefore extend it into HA-R2R, which contributes *16,844* curated instructions emphasizing social nuance, covering conversations, corridor crossings, and near-collision events. Table A3 presents representative directives, while Fig. A6 visualizes the expanded human-centric vocabulary.

We generate these enriched instructions via targeted LLM prompts (Appendix C.2), capturing diverse social scenarios. This augmentation shifts navigation from static paths to socially contingent routes, e.g., "avoid the couple chatting near the bar." Comparative analyses (Appendix C.3) highlight both the annotation workload and HA-R2R's potential for human-aware navigation.

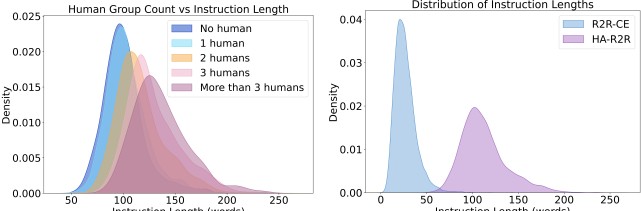

Figure 4: **HA-R2R Dataset Analysis. (a)** Distribution of instruction length by human group size (none to >3). **(b)** Comparison of instruction lengths between HA-R2R and R2R-CE.

**HA-VLN-VL Agent.** The HA-VLN-VL focuses on visual–language alignment. Adapted from Recurrent VLN-BERT Hong et al. (2021), it replaces actor–critic methods (e.g., A2C Konda & Tsitsiklis (1999)) with a streamlined imitation learning objective, isolating the contribution of multimodal grounding. At timestep $t$, the agent updates its hidden state $s_t$ and predicts an action distribution:

$$s_t, \; p_t^a \; = \; \text{HA-VLN-VL}(s_{t-1}, X, V_t), \tag{3}$$

where $X$ is the tokenized instruction (often referencing multiple humans) and $V_t$ encodes the fused RGB–depth view. A Transformer with a specialized state token attends jointly to visual and linguistic tokens, and final probabilities are derived via pooled attention:

$$p_t^a \; = \; \overline{\text{AveragePool}}_{s,v}^l. \tag{4}$$

Fine-tuned from Prevalent Hong et al. (2021) on HA-R2R, HA-VLN-VL demonstrates how stronger grounding alone benefits navigation under socially complex conditions (Appendix C.6).

**HA-VLN-CMA Agent.** HA-VLN-CMA emphasizes collision avoidance and real-time adaptation. Built on cross-modal attention (CMA) Krantz et al. (2020), it fuses textual embeddings $l = \text{BERT}(I)$ with visual features $v_t = \text{ResNet}(o_t)$. Multi-head attention produces a joint representation $f_t$, which an MLP maps to action probabilities:

$$P(a_t \mid f_t) \; = \; \text{Softmax}\big(\text{MLP}_{\text{action}}(f_t)\big). \tag{5}$$

Fig. A7(b) outlines the architecture (details in Appendix C.7). To address partial observability and unpredictable motion, we adopt Environmental Dropout (Envdrop) Tan et al. (2019) to simulate occlusions and Dataset Aggregation (DAgger) Ross et al. (2011) for iterative error correction. These strategies enhance re-planning when agents face obstacles or unexpected behaviors. Figs. A8 and 5 illustrate agent responses to bystanders, showing that collision risk and route deviation increase sharply in crowded passages. HA-VLN-CMA re-plans aggressively when blocked, whereas HA-VLN-VL leverages textual grounding to maintain appropriate distances. This contrast highlights our dual contributions: a socially enriched dataset (HA-R2R) and two baseline agents serving as extensible reference points. These baselines are not final solutions but starting points for the community to build, refine, and extend toward more advanced human-aware navigation models. Sec. 5 evaluates both agents on HA-VLN 2.0, demonstrating complementary strengths.

## 5 EXPERIMENTS

**Evaluation Metrics.** We evaluate performance on the HA-VLN 2.0 benchmark using two suites of metrics. **(1) Social compliance.** To assess social awareness, we use *Total Collision Rate* (TCR)

Table 1: **HA-VLN-CE Results Across Validation (Seen/Unseen) and Test Splits.** "HA-VLN-CMA*" denotes the full version of HA-VLN-CMA (+DA +EV). Metrics include NE (Navigation Error, meters), TCR (Total Collision Rate), CR (Collision Rate per step), and SR (Success Rate), with lower NE/TCR/CR and higher SR indicating better performance. All agents receive panoramic RGBD observations at each location.

| | Validation Seen | | | | | | | | Validation Unseen | | | | | | | | Test | | | | | | | |
| | Retrained | | | | Zero-shot | | | | Retrained | | | | Zero-shot | | | | Retrained | | | | Zero-shot | | | |
| Agent | NE↓ | TCR↓ | CR↓ | SR↑ | NE↓ | TCR↓ | CR↓ | SR↑ | NE↓ | TCR↓ | CR↓ | SR↑ | NE↓ | TCR↓ | CR↓ | SR↑ | NE↓ | TCR↓ | CR↓ | SR↑ | NE↓ | TCR↓ | CR↓ | SR↑ |
|---|---|---|---|---|---|---|---|---|---|---|---|---|---|---|---|---|---|---|---|---|---|---|---|---|
| HA-VLN-CMA-Base | 7.63 | 63.09 | 0.77 | 0.05 | 7.88 | 63.84 | 0.75 | 0.04 | 7.34 | 47.06 | 0.77 | 0.07 | 7.95 | 63.96 | 0.76 | 0.03 | 7.30 | 47.55 | 0.76 | 0.07 | 7.89 | 62.14 | 0.74 | 0.04 |
| HA-VLN-CMA-DA | 6.11 | 17.45 | 0.61 | 0.17 | 6.95 | 37.85 | 0.72 | 0.07 | 7.00 | 27.25 | 0.69 | 0.09 | 7.05 | 38.22 | 0.73 | 0.05 | 7.12 | 28.33 | 0.69 | 0.08 | 6.98 | 36.53 | 0.73 | 0.06 |
| HA-VLN-CMA* | 5.61 | 3.34 | 0.60 | 0.17 | 7.10 | 29.99 | 0.69 | 0.11 | 6.23 | 8.10 | 0.69 | 0.10 | 6.62 | 32.48 | 0.70 | 0.09 | 6.64 | 9.23 | 0.72 | 0.09 | 7.09 | 31.80 | 0.75 | 0.09 |
| HA-VLN-VL | 5.02 | 4.44 | 0.52 | 0.20 | 7.82 | 3.67 | 0.45 | 0.05 | 5.35 | 6.63 | 0.59 | 0.14 | 7.15 | 3.97 | 0.46 | 0.06 | 5.52 | 5.96 | 0.63 | 0.14 | 7.41 | 3.38 | 0.58 | 0.07 |
| BEVBert An et al. (2023) | 5.53 | 3.64 | 0.46 | 0.27 | 6.11 | 4.29 | 0.47 | 0.19 | 5.51 | 4.71 | 0.55 | 0.21 | 6.10 | 5.72 | 0.56 | 0.15 | 6.33 | 4.25 | 0.58 | 0.18 | 6.54 | 4.39 | 0.54 | 0.14 |
| ETPNav An et al. (2024) | 5.17 | 4.07 | 0.43 | 0.24 | 7.72 | 6.31 | 0.61 | 0.12 | 5.43 | 6.94 | 0.58 | 0.17 | 7.40 | 7.94 | 0.71 | 0.08 | 5.94 | 5.96 | 0.58 | 0.16 | 7.59 | 5.64 | 0.73 | 0.09 |

○ **Visited Node**    ● **Ghost Node**    ● **Final Goal**    ○ **Selected Goal**    ── **Traversed Path**

**Instruction:** Instruction: "Begin by ascending the steps in front of you. Upon reaching the top, make a left turn. As you proceed, you will enter a hallway. Be aware of a child who might be running through this area ... Continue moving forward until you reach the main room. Inside, you will observe a group of friends gathered around a table ... Remain at the doorway."

**Instruction:** "Instruction: "Begin your journey in the living room, where you will observe a person engaged in a quiet phone conversation. Move carefully to avoid disturbing them. Proceed through the hall and make a left turn. As you navigate ... Your endpoint is on the rug, positioned directly behind the desk chair."

Figure 5: **Agent Trajectory Examples (HA-VLN-CMA\*).** The top row demonstrates a failed navigation scenario where the agent fails to avoid an oncoming human, ultimately resulting in a collision. In contrast, the bottom row showcases a successful navigation: the agent proactively adjusts its trajectory to the left, effectively avoiding human interference and completing the task without collision.

and *Collision Rate* (CR). TCR measures the overall frequency of collisions, while CR reflects the proportion of socially inappropriate interactions. **(2) Navigation accuracy.** We report *Navigation Error* (NE) and *Success Rate* (SR). A trajectory is deemed successful under SR not only when the agent stops sufficiently close to the goal Anderson et al. (2018), but also when it demonstrates effective obstacle avoidance. Formal definitions of these metrics are provided in Appendix D.1.

We evaluate agents in two settings: **(1)** We assess the performance of HA-VLN 2.0 agents alongside several top agents on the **HA-VLN 2.0 benchmark**, utilizing both **HA-VLN-CE (continuous)** and **HA-VLN-DE (discrete)** (Sec. 5.1). We conduct extensive analysis and ablation studies examining key factors including continuous versus discrete settings, cross-domain generalization capabilities, human presence and interaction enrichment, step size variations, and sensor modality configurations. These analyses investigate their respective impacts on human-aware navigation performance and reveal complementary knowledge between the DE and CE approaches. **(2)** We deploy and evaluate HA-VLN 2.0 agents in real-world robotic scenarios across diverse layouts (office spaces, living rooms, hallways, and lobbies) with free-moving human volunteers (Sec. 5.2, Appendix D.5).

## 5.1 BENCHMARKING AGENTS ON HA-VLN 2.0

**HA-VLN-CE.** We systematically benchmark two notable continuous navigation agents, BEVBert An et al. (2023) and ETPNav An et al. (2024), together with our HA-VLN-CMA and HA-VLN-VL agents in Table 1. Each approach is trained/evaluated under two configurations: **Retrained**, where agents are trained/evaluated solely on HA-VLN-CE benchmark (HA-VLN-CE simulator + HA-R2R instruction dataset), and **Zero-shot**, where agents are trained solely on VLN-CE benchmark (VLN-CE simulator + R2R-CE) and evaluated on our benchmark. Table 1 shows pronounced gains when models incorporate HA-VLN-CE benchmark. For instance, BEVBert's SR

increases from 0.19 to 0.27 in seen split and from 0.15 to 0.21 in unseen. In stark contrast, Table 3 shows that BEVBert trained on our benchmark performs comparably to the VLN-CE-trained one on VLN-CE benchmark (SR: 0.35 vs. 0.37). This bidirectional evaluation suggests that explicit references to dynamic crowd behavior enhance real-world navigational readiness and confirm the robustness of HA-VLN-CE. Figure 5 presents navigation visualizations of HA-VLN-CMA* agent on the HA-VLN-CE benchmark, including one successful and one failed example. These examples demonstrate that dynamic human activities indeed increase the difficulty of navigation, while also making the scenarios more realistic and reflective of real-world challenges.

**HA-VLN-DE.** Table 2 compares top discrete agents on both VLN and HA-VLN-DE benchmarks, showing discrete agents can achieve moderate SR yet suffer high collisions in crowded scenes. For example, while Airbert Guhur et al. (2021) achieves a moderate SR at 0.36, it can incur a CR of up to 0.83, illustrating persistent collision risks. The results showcase adaptive collision-avoidance strategies also remain essential in discrete settings. Approaches that overlook human dynamics often fail when multiple bystanders converge (Sec. 3), particularly in tight junctions or doorways.

Table 2: **DE performance** of agents trained on VLN vs. HA-VLN-DE (Unseen). All agents use panoramic RGB observations.

| Agent | VLN | | HA-VLN-DE | | | |
|---|---|---|---|---|---|---|
| | NE↓ | SR↑ | NE↓ | TCR↓ | CR↓ | SR↑ |
| Speaker-Follower | 6.62 | 0.35 | 7.44 | 0.32 | **0.72** | 0.21 |
| Rec (PREVALENT) | **3.93** | **0.63** | 6.12 | 0.29 | 0.81 | 0.33 |
| Rec (OSCAR) | 4.29 | 0.59 | 6.12 | **0.28** | 0.78 | 0.34 |
| Airbert | 4.01 | 0.62 | **5.54** | 0.30 | 0.83 | **0.36** |
| NavCoT | 6.26 | 0.40 | 6.83 | 0.36 | 0.85 | 0.23 |

**Analysis & Ablation Studies.** *(1) Cross-domain Generalization.* Table 3 reveals that HA-R2R-trained agents achieve comparable SR to R2R-CE-trained agents (0.27 vs. 0.29) on R2R-CE validation set, while they outperform by +28.6% SR on the HA-R2R validation set, showcasing HA-R2R improves in-domain performance while maintaining cross-domain robustness. *(2) Human Presence and Interaction Enrichment.* Table 4 (a) shows in human presence ablations, replacing humans with cylinders drops TCR by around 36% and raises SR by around 10%, while removing human interaction enrichment drops TCR by up to 22% and raises SR by up to 25%, confirming humans are not merely treated as generic moving obstacles during navigation. *(3) Step Size.* Table 4 (b) indicates a degree of knowledge complementarity between DE and CE navigation when collisions are detected only at the endpoint of a step. Specifically, increasing the step size (from 0.1 m to 1.0 m), approximating DE-style navigation, can improve performance. We also conducted an additional experiment (Table A4) in which a 1.0 m step was treated as four 0.25 m sub-steps, and a 2.25 m step as nine 0.25 m sub-steps, with collisions checked after each sub-step. When evaluated on BEVBert in the val_unseen split, the agents failed to navigate effectively with both 1.0 m and 2.25 m step sizes, with SR dropping close to zero. These results highlight the need to account for the potentially "teleport-like" movement behaviors in DE when considering complementarity. *(4) Sensor Modalities.* Table 5 confirms that either adding depth or RGB consistently lowers collisions and raises SR, reflecting the importance of 3D cues for navigating around moving bystanders.

## 5.2 LEADERBOARD & REAL-WORLD VALIDATION

**HA-R2R Test Dataset & Leaderboard.** Building on R2R-CE, we present HA-R2R, featuring 16,844 instructions across 90 building scans with 910 annotated human models (see Secs. 3 & 4). While retaining path continuity from R2R-CE, we introduce refined goals to emphasize social awareness. The test partition of HA-R2R contains 3,408 instructions across 18 withheld buildings and intentionally emphasizes multi-human routes. To assess performance on this challenging test split, we host leaderboards for HA-R2R-DE and HA-R2R-CE benchmarks, evaluating both collision-related metrics (TCR, CR) and navigation metrics (NE, SR). We prepare an interactive interface shown in Figure 6 (a), where participants can explore the simulator from nine different views to examine all the annotated human motions and the surrounding environments. This allows them

Table 3: **Cross Domain Evaluation of BEVBert (CE) vs. Rec (PREVALENT) (DE).** Each model is trained/validated under different simulators (HA-VLN-CE/HA-VLN-DE vs. VLN-CE/VLN-DE) and different instruction sets (HA-R2R vs. R2R-CE/R2R). The blue cells () indicate performance changes when models are trained on R2R/R2R-CE instructions but validated on HA-R2R.

| Env | Training | | Validation | | Val (Unseen) | | Env | Training | | Validation | | Val (Unseen) | |
|---|---|---|---|---|---|---|---|---|---|---|---|---|---|
| | Simulator | Instr. | Simulator | Instr. | NE↓ | SR↑ | | Simulator | Instr. | Simulator | Instr. | NE↓ | SR↑ |
| CE | VLN-CE | R2R-CE | VLN-CE | R2R-CE | 4.57 | 0.37 | DE | VLN-DE | R2R | VLN-DE | R2R | 3.93 | 0.48 |
| | HA-VLN-CE | HA-R2R | | R2R-CE | 5.11 | 0.35 | | HA-VLN-DE | R2R | | R2R | 4.62 | 0.45 |
| | HA-VLN-CE | HA-R2R | HA-VLN-CE | HA-R2R | 4.35 | 0.27 | | HA-VLN-DE | HA-R2R | HA-VLN-DE | HA-R2R | 5.86 | 0.36 |
| | | R2R-CE | | R2R-CE | 4.13 | 0.29 | | | R2R | | R2R | 5.21 | 0.33 |
| | | HA-R2R | | HA-R2R | 5.51 | 0.21 | | | HA-R2R | | HA-R2R | 5.01 | 0.39 |
| | | R2R-CE | | HA-R2R | 6.23 (↑13.1%) | 0.15 (↓28.6%) | | | R2R | | HA-R2R | 6.11 (↑22.0%) | 0.24 (↓38.5%) |

Table 4: **Left: (a). Impact of Human Presence (hp) and Interaction Enrichment (enrich).** We evaluate without hp (replace human with cylinders) and without enrich (skip interaction & movement enrichment in Sec. 3, Appendix B.5) on both CE and DE settings. Rec (PRE) denotes Rec (PREVALENT). **Right: (b). Impact of Step Size on Navigation.** Here the collision is detected only at endpoint of a step, thus increasing step size transitions from finer-grained control to more discrete (teleport-potential) steps (default step size for CE is 0.25m). We show results for both **BEVBert** An et al. (2023) and **ETPNav** An et al. (2024) on seen/unseen.

| hp | enrich | Env | Agent | NE↓ | TCR↓ | CR↓ | SR↑ |
|----|--------|-----|-------|-----|------|-----|-----|
| ✓ | ✓ | CE | BEVBert | 6.10 | 5.72 | 0.56 | 0.15 |
| | | CE | ETPNav | 7.40 | 7.94 | 0.71 | 0.08 |
| | | DE | Rec (PRE) | 7.31 | 0.31 | 0.79 | 0.22 |
| ✓ | ✗ | CE | BEVBert | 6.32 (↑3.6%) | 5.11 (↓10.7%) | 0.46 (↓17.9%) | 0.17 (↑13.3%) |
| | | CE | ETPNav | 7.35 (↓0.6%) | 6.12 (↓22.9%) | 0.63 (↓11.3%) | 0.10 (↑25.0%) |
| | | DE | Rec (PRE) | 7.52 (↑2.9%) | 0.27 (↓12.9%) | 0.64 (↓19.0%) | 0.27 (↑22.7%) |
| ✗ | ✗ | CE | BEVBert | 6.13 (↑0.5%) | 3.25 (↓43.2%) | 0.35 (↓37.5%) | 0.19 (↑26.7%) |
| | | CE | ETPNav | 7.75 (↑4.7%) | 4.47 (↓43.7%) | 0.53 (↓25.4%) | 0.14 (↑75.0%) |
| | | DE | Rec (PRE) | 7.33 (↑0.3%) | 0.19 (↓38.7%) | 0.42 (↓46.8%) | 0.26 (↑18.2%) |

| Agent | Step Size | Validation (Seen) | | | | Validation (Unseen) | | | |
|-------|-----------|------|------|-----|-----|------|------|-----|-----|
| | | NE↓ | TCR↓ | CR↓ | SR↑ | NE↓ | TCR↓ | CR↓ | SR↑ |
| **BEVBert** | 0.10 | 5.65 | 8.43 | 0.50 | 0.23 | **5.41** | 12.60 | 0.54 | 0.22 |
| | 0.25 (CE Default) | **5.53** | 3.64 | 0.46 | 0.27 | 5.51 | 4.71 | 0.55 | 0.21 |
| | 0.40 | 5.60 | 1.77 | 0.39 | 0.28 | 5.63 | 2.63 | 0.44 | 0.25 |
| | 1.00 | 5.82 | 0.42 | 0.21 | **0.29** | 5.54 | 0.63 | 0.26 | **0.26** |
| | 2.25 | 7.66 | **0.09** | **0.10** | 0.03 | 7.23 | **0.10** | **0.10** | 0.03 |
| **ETPNav** | 0.10 | 5.15 | 11.70 | 0.54 | 0.20 | 5.47 | 18.66 | 0.64 | 0.16 |
| | 0.25 (CE Default) | 5.17 | 4.07 | 0.43 | 0.24 | 5.43 | 6.94 | 0.58 | 0.17 |
| | 0.40 | **5.11** | 2.43 | 0.36 | **0.26** | **5.32** | 3.77 | 0.46 | **0.21** |
| | 1.00 | 6.67 | 0.49 | 0.25 | 0.24 | 6.76 | 0.79 | 0.32 | 0.17 |
| | 2.25 | 7.61 | **0.10** | **0.10** | 0.02 | 7.21 | **0.13** | **0.12** | 0.03 |

Table 5: **Ablation on RGB/Depth Inputs.** We compare **BEVBert** An et al. (2023) and **ETPNav** An et al. (2024) on seen/unseen validations. ✓ denotes the sensor is enabled, while ✗ is disabled. Blue cells highlight performance changes (in %) upon removing/adding a modality. Best viewed in color.

| Agent | RGB | Depth | Validation (Seen) | | | | Validation (Unseen) | | | |
|-------|-----|-------|------|------|-----|-----|------|------|-----|-----|
| | | | NE↓ | TCR↓ | CR↓ | SR↑ | NE↓ | TCR↓ | CR↓ | SR↑ |
| **BEVBert** An et al. (2023) | ✓ | ✗ | 6.23 (↑12.6%) | 4.55 (↑25.0%) | 0.49 (↑6.5%) | 0.19 (↓29.6%) | 5.79 (↑5.1%) | 4.97 (↑5.5%) | 0.53 (↑3.6%) | 0.15 (↓28.6%) |
| | ✗ | ✓ | 5.68 (↑2.7%) | 3.77 (↑3.6%) | 0.48 (↑4.3%) | 0.25 (↓7.4%) | 5.50 (↓0.2%) | 4.73 (↑0.4%) | 0.53 (↑3.6%) | 0.20 (↓4.8%) |
| | ✓ | ✓ | 5.53 | 3.64 | 0.46 | 0.27 | 5.51 | 4.71 | 0.55 | 0.21 |
| **ETPNav** An et al. (2024) | ✓ | ✗ | 6.14 (↑18.8%) | 6.07 (↑49.1%) | 0.56 (↑30.2%) | 0.17 (↓29.2%) | 6.38 (↑17.5%) | 7.44 (↑7.2%) | 0.65 (↑12.1%) | 0.13 (↓23.5%) |
| | ✗ | ✓ | 4.92 (↓4.8%) | 5.45 (↑33.9%) | 0.55 (↑27.9%) | 0.21 (↓12.5%) | 5.94 (↑9.4%) | 7.23 (↑4.2%) | 0.65 (↑12.1%) | 0.16 (↓5.9%) |
| | ✓ | ✓ | 5.17 | 4.07 | 0.43 | 0.24 | 5.43 | 6.94 | 0.58 | 0.17 |

Figure 6: **(a). Interactive interface** we provide to explore 910 annotated human models and environments in HA-VLN 2.0 simulator from nine views. **(b). Human-aware navigation with multiple bystanders.** *Left:* Instruction provided to the robot. *Right:* A third-person view illustrates the robot's trajectory among dynamic bystanders, and selected robot observations from onboard camera.

to gain a deeper understanding of the challenging dynamic scenarios we provide. Submissions may include agent code or trajectories, providing reproducible, server-side evaluations and setting a new benchmark for human-centric, dynamic VLN research.

**Real-World Validation & Setup.** We deploy our trained agents on a *Unitree Go2-EDU* quadruped, equipped with Intel Realsense D435i RGB-D camera, MID360 3D LiDAR, and IMU for onboard perception and control. As Figure 6 (b) illustrates, experiments are conducted in four indoor spaces (office, living room, hallway, lobby), each populated by 2–4 free-moving volunteers. Implementation details and more visual examples are provided in Appendix D.5. The agent navigates safely in moderately congested conditions but faces challenges in tight corridors or sudden crowd convergence, highlighting the need for robust re-planning under partial observability.

# 6 CONCLUSION

We presented *HA-VLN 2.0*, a unified framework that standardizes discrete and continuous VLN under explicit human-centric constraints. By integrating dynamic human motion, refined annotations, and high-fidelity simulators, our *HA-R2R* dataset emphasizes human-centric instructions. Experiments show social awareness, multi-human interactions, and partial observability greatly increase complexity, reducing advanced agents' performance. Nevertheless, our approach balances safety, efficiency, and personal space. Real-world tests confirm sim-to-real transfer, while our public leaderboard standardizes evaluations. By releasing all data, simulators, agents, and tools, we promote socially responsible, context-aware navigation in dynamic, human-populated environments.

## Ethics Statement

This work adheres to the ICLR Code of Ethics. In this study, no human subjects or animal experimentation was involved. All datasets used, including HAPS 2.0, HA-R2R, HA-VLN simulator annotation data, were sourced in compliance with relevant usage guidelines, ensuring no violation of privacy. We have taken care to avoid any biases or discriminatory outcomes in our research process. No personally identifiable information was used, and no experiments were conducted that could raise privacy or security concerns. We are committed to maintaining transparency and integrity throughout the research process.

## Reproducibility Statement

We have made every effort to ensure that the results presented in this paper are reproducible. The task, simulator and agent setup, including annotation details, simulator functions, and model configurations, are described in detail in the paper. We have also provided a full description of the simulator and dataset setup in the supplementary to assist others in reproducing our experiments.

Additionally, HAPS 2.0, HA-R2R, HA-VLN simulator annotation data are available, ensuring consistent and reproducible evaluation results. We believe these measures will enable other researchers to reproduce our work and further advance the field.

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

# Appendix

This supplementary material provides expanded details and results that complement the main paper. Section A offers a comprehensive literature survey focusing on three key research challenges. Section B describes our dataset construction, annotation protocols, real-time rendering methods, API design, and additional insights on annotation data. Section C presents an in-depth overview of the HA-R2R dataset and the proposed navigation agents. Finally, Section D includes detailed evaluation metrics, additional numerical results, visualized navigation outcomes, and real-world robot validation studies, each supplemented with thorough analysis. For further resources, access project page https://ha-vln-webpage.vercel.app/, and data samples provided in supplementary.

## A    RELATED WORK

This appendix surveys the evolution of Vision-and-Language Navigation (VLN) tasks, simulators, and agent designs, with particular attention to how *Human-Aware VLN (HA-VLN) 2.0* advances the state of the art. We focus on three key aspects deemed critical for bridging the Sim2Real gap: *(1) Socially Compliant Navigation*, *(2) Human-Aligned Instructions and Visual Cues*, and *(3) Dynamic Environments with Human Activities and Interactions*. Table A1 summarizes how prior work compares under these dimensions.

### A.1    DEVELOPMENT OF VLN TASKS

Early VLN tasks focused on basic indoor navigation—exemplified by Room-to-Room (R2R) Anderson et al. (2018); Fried et al. (2018); Gu et al. (2022); Ku et al. (2020)—and outdoor tasks like TOUCHDOWN Chen et al. (2019) and MARCO MacMahon et al. (2006). Later efforts such as REVERIE Qi et al. (2020) and VNLA Nguyen et al. (2019) introduced object-centric or goal-driven navigation. While these approaches expanded the range of tasks, they typically overlooked real human behavior and social contexts. Dialogue-based tasks (e.g., DialFRED Gao et al. (2022), CVDN Thomason et al. (2020)) incorporated interactive elements but did not account for dynamically moving bystanders or social-distance constraints. Initiatives like VLN-CE Krantz et al. (2020) moved closer to real-world conditions by enabling continuous navigation, yet remained devoid of explicit human factors Jain et al. (2019); Ku et al. (2020); Nguyen et al. (2019); Thomason et al.

(2020). HA3D Li et al. (2024) addressed human motion and included human-oriented instructions but did not require agents to conform to social norms—e.g., maintaining safe distances or refraining from disturbing ongoing activities. Our proposed *HA-VLN 2.0* addresses these gaps by embedding all three essential elements, socially compliant navigation, human-referenced instructions, and dynamic human activities, into a single framework. Agents must plan routes among unpredictable bystanders, interpret language mentioning people and their behaviors, and uphold social standards. This integrated setup results in a benchmark that closely aligns with real-world navigation demands.

## A.2 SIMULATORS FOR VLN TASKS

A reliable simulator is essential for developing and evaluating VLN agents. Early simulators like Matterport3D Anderson et al. (2018) and House3D Wu et al. (2018) offered photorealistic or synthetic indoor environments but lacked mobile humans. Others, such as AI2-THOR Kolve et al. (2017) and Gibson Xia et al. (2018), introduced more interactive elements yet typically assumed static or purely synthetic contexts, thus limiting their applicability for studying social compliance. Google Street View, used in some outdoor navigation tasks, presents static imagery with occasional humans in the scene but lacks dynamic or interactive elements. HA3D Li et al. (2024) moved a step further by including human activities and instructions referencing people, though it did not mandate socially compliant navigation. HabiCrowd Vuong et al. (2024) integrated crowds into photorealistic domains, improving visual diversity but omitting human-aligned instructions. Similarly, Habitat 3.0 Savva et al. (2019) provides high-performance simulation without extensive multi-human or social-compliance features. By contrast, our *HA-VLN Simulator* unifies dynamic human activities, photorealistic rendering, and social-compliance requirements. Agents perceive and react to evolving bystander behaviors—such as avoiding collisions or maintaining personal space—using both discrete and continuous navigation. Specifically, we introduce 675 scenes (across 90 scenarios), 122 motion types, and a cohesive framework that supports instruction-driven dynamic human interactions. By supporting both discrete and continuous action spaces, HA-VLN further broadens its potential for addressing diverse navigation goals and real-world deployment challenges.

## A.3 AGENTS FOR VLN TASKS

From early attention-based and reinforcement-learning approaches Ma et al. (2019); Qi et al. (2020); Wang et al. (2019) to modern vision-language pre-training Lu et al. (2019); Hao et al. (2020); Li et al. (2020), VLN agents have grown increasingly adept at parsing instructions and navigating complex environments. However, most existing solutions, including EnvDrop Tan et al. (2019), PREVA-LENT Hao et al. (2020) and VLN-BERT Hong et al. (2021), rely on panoramic navigation, streamlining the action space but limiting realism of their movement. Recent efforts like NavGPT Zhou et al. (2024) and NaVid Zhang et al. (2024a) explore continuous, egocentric navigation in partially dynamic worlds, yet they still lack explicit attention to *human-aligned* instructions or *social compliance*. In particular, these agents may not recognize the need to maintain safe distances, avoid disturbing activities, or adapt routes with active bystanders. HA-VLN agents address these gaps by navigating among multiple, moving humans and adhering to social norms. They interpret fine-grained, human-centric instructions and leverage visual cues that reflect real-world interactions, ensuring collision-free, respectful travel. This fusion of social compliance and human dynamics sets HA-VLN apart, aligning agent behavior more closely with real-world challenges Dong et al. (2025).

## B SIMULATOR DETAILS

### B.1 HAPS DATASET 2.0

We develop HAPS 2.0 to address the shortcomings of its predecessor Li et al. (2024), particularly in terms of mismatches between textual descriptions and motion data, as well as the limited diversity of region–motion associations.

**Motion–Description Alignment.** The original HAPS dataset contains 435 motion categories, each defined by a region (e.g., *hallway*) and a textual description (e.g., "Someone talking on the phone while pacing"). However, more than half of these pairs do not match accurately. We therefore conduct a two-round manual verification, where multiple volunteers determine whether each pair is valid. Motions that fail both rounds are removed, yielding 172 precisely aligned motions.

Table A1: Comparison of VLN tasks, simulators, and agents based on *(1) Socially Compliant Navigation*, *(2) Human-aligned Instructions and Visual Cues*, and *(3) Dynamic Environments with Human Activities*.

| | Socially Compliant Navigation | Human-aligned Instructions and Visual Cues | Dynamic Environments | Prior Work |
|---|---|---|---|---|
| **Tasks** | ✗ | ✗ | ✗ | MARCO MacMahon et al. (2006), DRIF Blukis et al. (2018), VLN-R2R Anderson et al. (2018), TOUCHDOWN Chen et al. (2019), REVERIE Qi et al. (2020), Dial-FRED Gao et al. (2022) VNLA Nguyen et al. (2019), CVDN Thomason et al. (2020), R4R Jain et al. (2019), RxR Ku et al. (2020), EQA Das et al. (2018), IQA Gordon et al. (2018) |
| | ✗ | ✗ | ✓ | VLN-CE Krantz et al. (2020) |
| | ✗ | ✓ | ✓ | HA3D Li et al. (2024) |
| | ✓ | ✓ | ✓ | **HA-VLN (Ours)** |
| **Simulators** | ✗ | ✗ | ✗ | Matterport3D Anderson et al. (2018), House3D Wu et al. (2018), AI2-THOR Kolve et al. (2017), Gibson GANI Xia et al. (2018) |
| | ✗ | ✗ | ✓ | Habitat Savva et al. (2019), Google Street, ViZDoom Kempka et al. (2016) |
| | ✗ | ✓ | ✓ | HA3D Li et al. (2024) |
| | ✓ | ✓ | ✓ | **HA-VLN (Ours)**, Habitat3.0 Puig et al. (2023) |
| **Agents** | ✗ | ✗ | ✗ | EnvDrop Tan et al. (2019), AuxRN Zhu et al. (2020), PREVALENT Hao et al. (2020), RelGraph Hong et al. (2020), HAMT Chen et al. (2021), NavCoT Lin et al. (2025) Rec-VLNBERT Hong et al. (2021), EnvEdit Li et al. (2022), Airbert Guhur et al. (2021), Lily Lin et al. (2023), ScaleVLN Wang et al. (2023) |
| | ✓ | ✗ | ✓ | NavGPT Zhou et al. (2024), NaVid Zhang et al. (2024a), Student Force Anderson et al. (2018) |
| | ✓ | ✓ | ✓ | **HA-VLN Agent (Ours)** |

Table A2: **Comparison of HAPS 1.0 vs. HAPS 2.0.** We show the total number of motion categories, average *accuracy* and *compatibility* scores (both on a 1–10 scale), the number of failure cases (e.g., severe motion-description mismatches), and total annotation time. HAPS 2.0 features more diverse motions, improved motion-env alignment, and reduced failures, albeit at higher annotation effort.

| Datasets | Motions ↑ | Accuracy (1-10) ↑ | Compatibility (1-10) ↑ | Failure Cases ↓ | Annotation Time (hours) |
|---|---|---|---|---|---|
| HAPS 1.0 Li et al. (2024) | 435 | 6.3 | 5.9 | 120 | 320 (verified by Li et al. (2024)) |
| HAPS 2.0 (ours) | 486 | 8.5 | 8.1 | 0 | 430+ |

**Diversifying Region–Motion Relationships.** In the initial dataset, each region was tied to only a few rigidly defined motions (e.g., *hallway* mostly involves "pacing on a phone," *stairs* focuses on "sliding down a banister" or "decorating the stairway"). Such narrow mappings cause biases and limit the realism of agent navigation. To remedy this, we reorganize region–motion associations, adapting the same motion to fit various environments, including both indoor and outdoor scenes. For instance, "talking on the phone" is re-contextualized to reflect whether someone is pacing upstairs or moving around a meeting room. This broader approach offers more faithful representations of human behavior and reduces environmental biases, thus improving real-world applicability.

**HAPS 2.0 vs. HAPS 1.0.** Table A2 quantitatively contrasts HAPS 2.0 with HAPS 1.0. We recruit 26 volunteers to evaluate every motion in both datasets on two 1–10 scales (*motion accuracy*, *motion–environment compatibility*). A motion is deemed a failure if it scores under 3 in either category or below 5 in both. As shown, HAPS 2.0 achieves higher accuracy (8.5 vs. 6.3), better compatibility (8.1 vs. 5.9), and zero failures (0 vs. 120). It also increases motion diversity (486 vs. 435) and overall annotation effort (430+ vs. 320 hours). Moreover, HAPS 2.0 refines annotation workflows and simulator design for enhanced generalization.

Altogether, HAPS 2.0 includes 26 distinct regions across 90 architectural scenes, covering 486 human activities in both indoor and outdoor contexts. Fig. A2 illustrates these improvements. By offering more accurate, flexible, and diverse depictions of human actions, HAPS 2.0 provides a robust foundation for research in human motion analysis, social navigation, and beyond.

## B.2 COARSE ANNOTATION USING PSO

We adopt a coarse-to-fine strategy for positioning human motions in 3D scans. Initially, we define each region by boundary coordinates $\mathbf{B}_{\text{lo}} = (x_{\text{lo}}, y_{\text{lo}}, z_{\text{lo}})$, $\mathbf{B}_{\text{hi}} = (x_{\text{hi}}, y_{\text{hi}}, z_{\text{hi}})$, and compile an object list $\mathbf{O} = \{j_1, j_2, \ldots, j_n\}$ with positions $\mathbf{p}^{j_i}$. We then use Particle Swarm Optimization (PSO) Kennedy & Eberhart (1995) (more details are provided in Algorithm A1) to locate each motion $h_i$ at an optimal position $\mathbf{p}^{opt}$.

**Safe Distance Constraint.** We set $\epsilon = 1\,\text{m}$ as the minimum clearance between humans and objects, ensuring a realistic layout while leaving space for agent passage.

---

**Algorithm A1** Coarse Annotation via PSO

---

**Require:** Region $\mathbf{R} \leftarrow \langle \mathbf{r}, \mathbf{B}_{lo}, \mathbf{B}_{hi} \rangle$, where $\mathbf{r}$ is region label and boundary coordinates $\mathbf{B}_{lo} = (x_{lo}, y_{lo}, z_{lo})$ and $\mathbf{B}_{hi} = (x_{hi}, y_{hi}, z_{hi})$; object list $\mathbf{O} \leftarrow \{j_1, j_2, \ldots, j_n\}$ with positions $\mathbf{p}_{j_i} \leftarrow (x_{j_i}, y_{j_i}, z_{j_i})$; human motion set $\mathbf{H}$; minimum safe distance $\epsilon \leftarrow 1$ m; height offset $\Delta_z \leftarrow 0.75$ m.

**Ensure:** Final positions $\mathbf{p}^h \leftarrow (x_h, y_h, z_h)$ for each human motion $h \in \mathbf{H}$.

1: **while** not all human motions placed **do**
2:     Filter human motions $\mathbf{H}' \subseteq \mathbf{H}$ matching $\mathbf{r}$;
3:     Match objects $\mathbf{O}$ with human motions $\mathbf{H}'$ based on semantic similarity to form pairs $(h_i, j_i)$;
4:     **for** each pair $(h_i, j_i)$ **do**
5:         Define search space $\mathbf{S} \leftarrow \langle x_{lo}, x_{hi} \rangle \times \langle z_{lo}, z_{hi} \rangle \times \langle y_{lo}, y_{hi} \rangle$ around object $j_i$;
6:         Initialize PSO with particles randomly positioned within $\mathbf{S}$;
7:         Convergence criteria $\leftarrow$ minimal fitness change;
8:         **repeat**
9:             **for** each particle $p$ in the swarm **do**
10:                 Compute position $\mathbf{p}^h$ of particle $p$;
11:                 Compute fitness $f(p)$;
12:                 $f(p) \leftarrow d(\mathbf{p}^h, \mathbf{p}^{j_i}) + P_{\text{constraints}}(p)$;
13:                 where $d(\mathbf{p}^h, \mathbf{p}^{j_i})$ is the Euclidean distance, and $P_{\text{constraints}}(p)$ is the penalty for constraint violations;
14:                 **Constraints**:
15:                 $d(\mathbf{p}^h, \mathbf{p}^{j_i}) \leq 1$ m;           (Proximity to target object)
16:                 $d(\mathbf{p}^h, \mathbf{p}^{j_u}) \geq \epsilon, \forall j_u \in \mathbf{O}, j_u \neq j_i$;    (Maintain safe distance from other objects)
17:                 $\mathbf{p}^h \in \mathbf{R}$;                 (Within region boundaries)
18:                 Optional: $z_h \geq z_{j_i} + \Delta_z$;          (Height offset)
19:             **end for**
20:             Update particle velocities and positions using PSO update equations;
21:         **until** convergence criteria met
22:         Assign best particle position $\mathbf{p}^h$ to $h_i$;
23:         **if** no feasible solution found **then**
24:             Adjust PSO parameters and retry;
25:         **end if**
26:     **end for**
27: **end while**

---

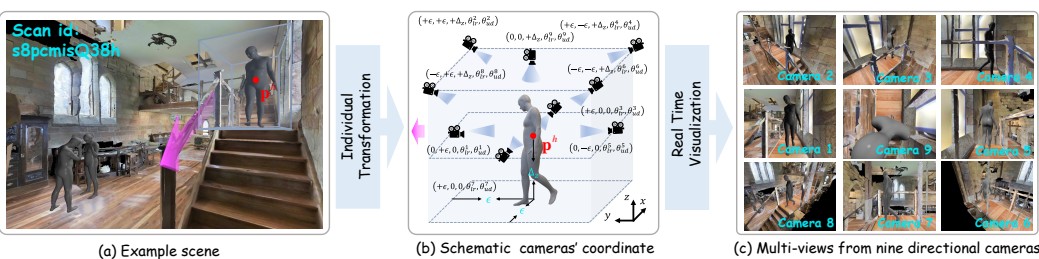

(a) Example scene        (b) Schematic cameras' coordinate      (c) Multi-views from nine directional cameras

Figure A1: **Multi-View Camera Setup. (a)** A sample scene overview. **(b)** Schematic illustrating the nine camera placements around the human figure, noting key coordinates and rotations. **(c)** Example snapshots from the nine directional cameras, each providing a distinct viewpoint for accurate motion annotation.

**Adaptive Penalties.** Our fitness function applies penalties to placements that violate constraints (e.g., intersecting walls or overlapping humans). This strategy discourages infeasible poses and promotes plausible scene geometry alignments. The resulting coarse alignment establishes a starting point, after which we apply finer manual or semi-automated adjustments to refine multi-human interactions and ensure consistent coverage of diverse motion types.

## B.3 FINE ANNOTATION USING A MULTI-CAMERA SETUP

To refine the coarse placements of human motions, we draw inspiration from 3D skeleton-capture methods Ji et al. (2018); Petrovich et al. (2021) and deploy nine RGB cameras, each positioned

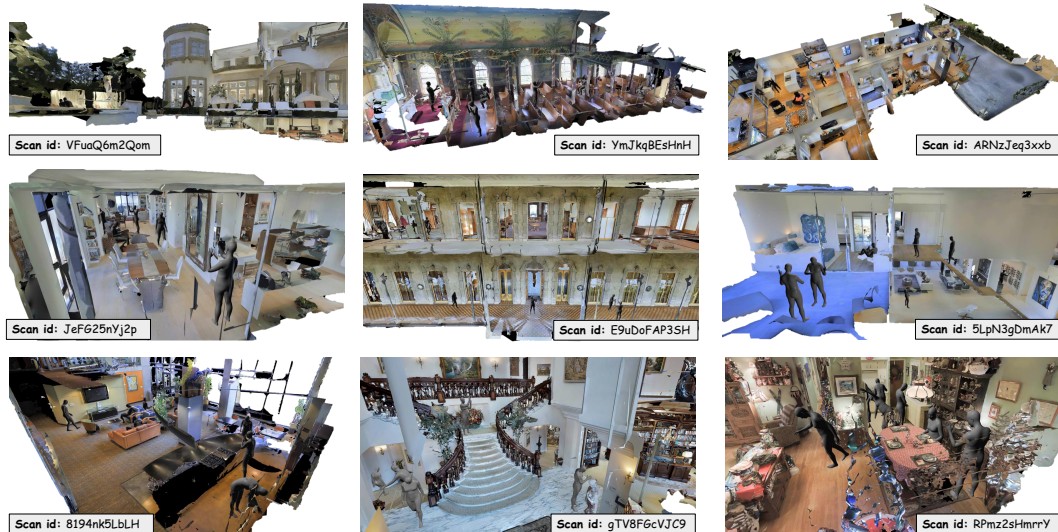

Figure A2: **Overview of HA-VLN Scenes.** These examples illustrate annotated human subjects across multiple scans in the HA-VLN simulator, highlighting a range of well-aligned motions, movements, and interactions (both with objects and with other humans).

around the motion site. As shown in Fig. A1, this arrangement provides a comprehensive multi-view perspective, revealing potential collisions or misalignments between the human figure and surrounding objects.

**Camera Positions & Angles.** For each camera $i$ ($i = 1, 2, \ldots, 8$), we set its 3D location $\mathbf{p}^{\text{cam}}$ to shift by $\Delta_x$, $\Delta_y$, and $\Delta_z$ from the base position $\mathbf{p}^{\text{h}}$. Horizontal rotation $\theta_{\text{lr}}^i$ is uniformly spaced at $\frac{\pi i}{8}$, while vertical rotation $\theta_{\text{ud}}^i$ depends on whether $i$ is odd or even:

$$\tan \theta_{\text{ud}}^i = \begin{cases} 0, & \text{if } i \text{ is odd,} \\ \frac{\Delta_z}{\sqrt{2}\,\epsilon}, & \text{if } i \text{ is even.} \end{cases} \tag{A1}$$

For the ninth camera (overhead view), $\theta_{\text{lr}}^9 = 0$ and $\theta_{\text{ud}}^9 = \frac{\pi}{2}$. These settings are ideal for general views and can be further adjusted in constrained spaces (e.g., narrow closets) or scenes requiring specialized viewpoints.

### B.4 FINE ANNOTATION PROTOCOL

We adopt the following six-step procedure to fine-tune a human's position and orientation:

1. *Initial View.* Generate an overall preview of the human figure at $\mathbf{p}^{\text{h}}$ (Fig. A1(a)).

2. *Multi-Camera Observations.* Collect images from the nine cameras (Figs. A1(b)–(c)). Adjust camera angles or offsets as necessary, particularly in tight scenes like small bathrooms or closets.

3. *Vertical Collision Checks.* Inspect overhead Camera 9 to detect vertical overlaps (e.g., arms interpenetrating a table). If collisions exist, identify the nearest side camera to determine how best to shift the figure.

4. *Horizontal Translation.* Modify $\Delta_x$ and $\Delta_y$ accordingly—if a nearby camera (e.g., Camera 1) reveals front-facing overlaps, shift $\mathbf{p}^{\text{h}}$ by adding or subtracting based on Camera 1's perspective.

5. *Side Cameras Review.* Examine Cameras 2–8 to catch lingering overhang or collisions. Adjust the figure's position proportionally, typically referencing a standard human height of 1.5 m to gauge whether shifts remain plausible.

6. *Finalize Output.* Upon confirming a collision-free layout, automatically generate the final video render and corresponding JSON metadata files.

This multi-camera process systematically eliminates misalignments, ensuring each human model remains properly integrated within the environment. The result is a more realistic portrayal of multi-human interactions and improved fidelity for downstream tasks.

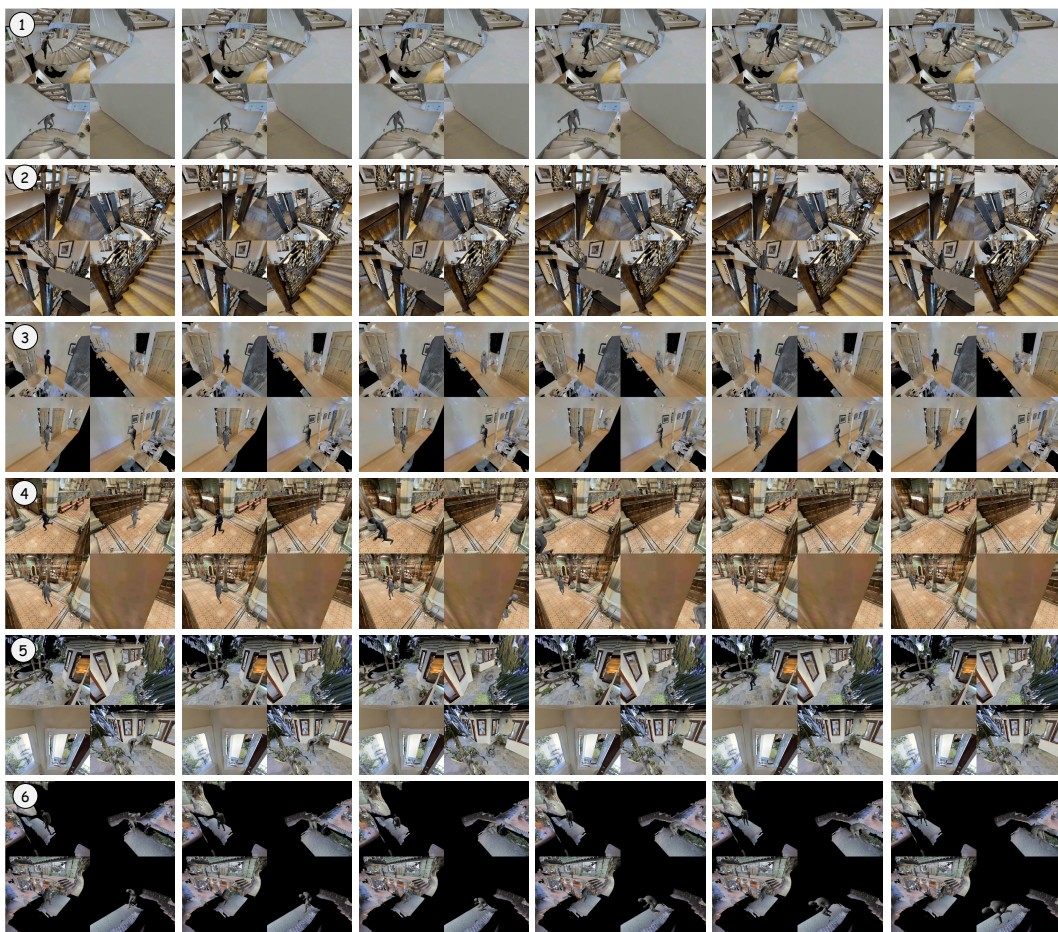

Figure A3: **Movement Examples.** We present representative frames from a single set of human motions, each annotated with its corresponding movement. Activities include ascending stairs, running, and pacing. For clarity, we highlight four camera views (Cameras 2, 4, 6, 8) within the multi-camera setup to provide a comprehensive perspective of human behaviors. *(Zoom in for finer details.)*

### B.5 MULTI-HUMAN INTERACTION & MOVEMENT ENRICHMENT

To diversify scenes and amplify interactivity, we place additional characters into regions already featuring human motion annotations. This enables more complex interactions and varied motion trajectories. Manual insertion of extra characters, however, is time-consuming and prone to subjective bias, limiting data reliability and diversity.

**Human-in-the-Loop Method.** We employ large language models (LLMs) such as ChatGPT-4 and LLaMA-3-8B-Instruct to propose plausible multi-human scenarios. Each prompt integrates details about existing human motions, object positions, and regional context, guiding the LLMs to generate rich, multi-character interactions. Our prompt design uses a *system prompt* and *few-shot examples* (Listings 1 and 2) to ensure clarity and detail. For instance, we collect each human's position and identify objects within 6 m, describing relative distances and orientations. The LLMs then construct additional human activities suited to the scene, merging them into cohesive multi-person narratives.

**Iterative Annotation Workflow.** After the LLMs produce candidate interactions, we merge outputs from ChatGPT-4 and LLaMA-3-8B-Instruct, then manually refine and validate them over four rounds Ding et al. (2024); Cheng et al. (2024). This process corrects inconsistencies and ensures contextual alignment. We subsequently place new human motions according to the generated descriptions, leveraging our multi-camera technique (Sec. B.3) for precise annotation of complex activities (e.g., stair-walking, see Fig. A3).

```
"system": "You are an expert in 3
    D scene understanding,
    specializing in generating
    detailed multi-person
    interaction scenarios in
    complex environments. I will
    provide you with information
    about the current human
    motion, object positions, and
     the region context. Your
    task is to infer possible
    additional human interactions
     and describe their actions
    in relation to the objects
    and scene."

"example": ...
//examples in Listing 2

"user": Input:
[human_motion]: \"...\";
[Region]: \"...\";
[Object_1]: \"... \";
[Object_2]: \"...\";
[Object_3]: \"...\";

Based on these inputs, provide a
    detailed, objective
    description of additional
    people in the scene, their
    locations, and their
    interactions with objects and
     each other. Ensure the
    interactions are plausible
    based on the given layout and
     actions.

Output: ...  //generation
```

Listing 1: Format of our LLM prompt for multi-human interactions & movements enrichment.

```
# Example 1
Input:
[human_motion]: "tv:An individual
    enjoying a sports game, cheering
    loudly.\"
[region]: "\tv.\"
[TV]: "\4 meters in front of the
    couch, placed on a white media
    console against the wall.\"
[couch]: "\Against the wall, 4
    meters away facing the TV.\"
[table]: "\1.5 meters in front of
    the couch, holding some snacks
    and drinks.\"

Output:
"Three friends are gathered in the
    living room, watching a movie
    together. Two of them are seated
    comfortably on the couch, which
    is positioned against the wall,
    facing the TV. They are sharing
    popcorn and drinks placed on a
    table 1.5 meters in front of the
    couch. The atmosphere is relaxed,
     with one friend fully focused on
     the movie, casually eating
    popcorn. Another friend seated
    next to him. In front of the TV,
    a third friend stands closer,
    about 2 meters from the couch, is
     more animated, loudly cheering
    as they switch their attention to
     a sports game playing on a
    different screen."
```

Listing 2: Few-shot example for for multi-human interactions & movements enrichment.

**Examples of Enriched Interactions.** Fig. 2 demonstrates how additional humans can populate a living room: *"two people sit on the couch, sharing popcorn on a small table,"* while *"a third friend stands in front of the TV, cheering."* Such enriched scenes capture realistic multi-human behaviors—from casual gatherings to active cheering—offering agents a broader range of social cues for navigation and interaction.

### B.6 REAL-TIME HUMAN RENDERING

We integrate dynamic human models into the simulation through a multi-threaded pipeline inspired by *Producer–Consumer* principles and Java-style signaling (check details in Algorithm A2). This setup enables agents to observe and respond to human motions in real time, facilitating adaptable navigation policies.

**System Initialization.** We begin by loading the environment $\mathcal{E}$, the set of human motions $\mathbf{H}$, and an object template manager $\mathcal{T}$ that handles 3D model templates efficiently.

---

**Algorithm A2** Real-time Human Rendering in Simulation

---

**Require:** Simulation environment $\mathcal{E}$; Human motion data $\mathbf{H}$; Signal queue $\mathcal{Q}$ with maximum size $M \leftarrow 120$; Total frames $N \leftarrow 120$; Frame interval $\Delta t$.
**Ensure:** Continuous real-time rendering of $\mathbf{H}$ within $\mathcal{E}$.
 1: Initialize simulator $\mathcal{E}$, object template manager $\mathcal{T}$ in $\mathcal{E}$, human motion data $\mathbf{H}$ and signal queue $\mathcal{Q}$;
 2: Initialize total signals sent and processed to 0;
 3: **// Thread 1: Signal sender thread**
 4: **while** true **do**
 5:    **if** not $\mathcal{Q}.\texttt{full()}$ **then**
 6:       Enqueue signal "REFRESH_HUMAN" into $\mathcal{Q}$;
 7:       Increment total signals sent;
 8:    **end if**
 9:    Sleep for $\Delta t$;
10: **end while**
11: **// Thread 2: Main thread**
12: **while** simulation is running **do**
13:    **if** new episode starts **then**
14:       Clear $\mathcal{Q}$ and reset total signals sent to 0;
15:       Remove previous human models from $\mathcal{E}$;
16:    **end if**
17:    **// Agent handles signals before observation**
18:    **while** not $\mathcal{Q}.\texttt{empty()}$ **do**
19:       Dequeue signal from $\mathcal{Q}$;
20:       $t \leftarrow$ (total signals processed) $\mathrm{mod}\ N$ {Compute current frame index};
21:       Remove previous human models from $\mathcal{E}$;
22:       **for** each human motion $h \in \mathbf{H}$ **do**
23:          Retrieve motion category, translation, and rotation of $h$ at frame $t$;
24:          Load template $\tau_h$ into $\mathcal{T}$;
25:          Add human $o_h$ to $\mathcal{E}$ using template $\tau_h$;
26:          Set translation and rotation of $o_h$;
27:       **end for**
28:       Increment total signals processed;
29:    **end while**
30:    Agent observes environment and makes decision;
31: **end while**

---

**Signal Sender Thread (Thread 1).** At intervals $\Delta t$, Thread 1 places "refresh" signals into a queue $\mathcal{Q}$. If $\mathcal{Q}$ is full, it pauses until earlier signals are processed, preventing data overload. This thread models a continuous stream of human motion updates at a fixed frequency.

**Main Simulation Thread (Thread 2).** When the agent is about to act, Thread 2 checks $\mathcal{Q}$ for pending refresh signals. It calculates the current frame index $t$ as (signals_processed $\mathrm{mod}\ N$), where $N$ is the total length of the human motion sequence. Template manager $\mathcal{T}$ then removes outdated models and loads frame $t$ into the environment, adjusting each figure's position and orientation.

**Synchronization & Consistency.** We refresh human models immediately before the agent's perception step, ensuring it observes the latest motion state. Upon starting a new episode, $\mathcal{Q}$ is cleared, and signal counters reset, so human motions revert to frame 0, maintaining consistency across episodes. This real-time process keeps human activities synchronized with agent's action cycle, creating dynamic scenes where agents must adapt to changing bystander locations and behaviors.

B.7 API DESIGN

**Discrete Environment (DE).** In our discrete setting, all agent and human positions are tracked via a real-time navigational graph displayed in a 2D top-down view. Each human's activity is stored as a tuple $\langle p_h, d_{agent}, \theta_{relative}, a_{status} \rangle$, where $p_h$ is the human's 2D coordinate, $d_{agent}$ is the distance to the agent, $\theta_{relative}$ is the relative orientation, and $a_{status}$ indicates activity state. This representation supports efficient, simultaneous tracking of multiple humans in a discrete viewpoint space.

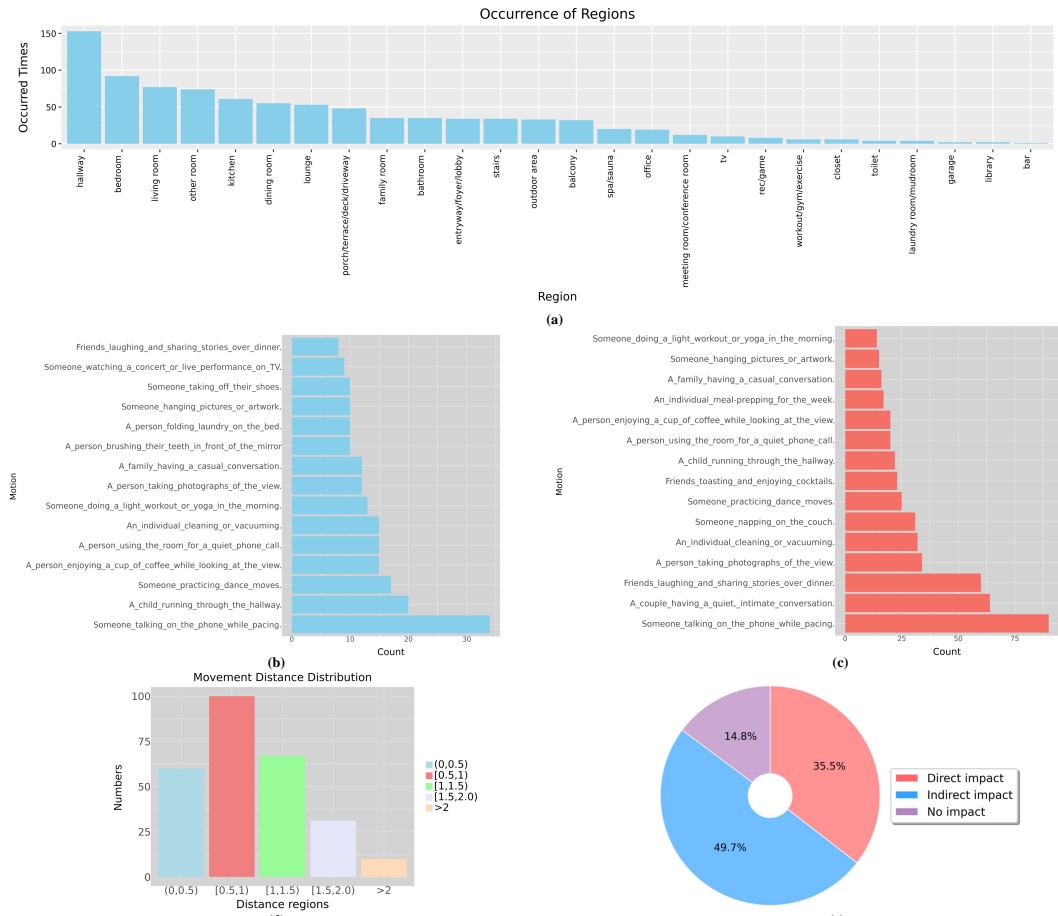

Figure A4: Statistics on human annotations in HA-VLN simulator: (a) Distribution of humans by 26 region types; (b) Top 15 motions without multi-human interaction & movement enrichment; (c) Top 15 motions with enrichment; (d) Distribution of human trajectory lengths (in meters); (e) Impact of human presence on environment, categorized as direct, indirect, and no impact. (Zoom in to view)

*Multi-Entity Detection & Tracking.* We employ object detection on each discrete panorama to identify humans, assigning unique IDs for continuous monitoring throughout the navigation process. By linking recognized human poses to specific graph nodes, we anchor their activities to well-defined spatial references.

*User Interface.* A specialized UI presents a bird's-eye view of the 2D graph, allowing researchers to visualize, annotate, and adjust human behaviors in real time. This interface significantly streamlines data annotation and analysis for discrete human-aware navigation research.

**Continuous Environment (CE).** Our API in continuous mode mainly focuses on three components: *(1) Human Activity Monitoring*, *(2) Environmental Perception*, and *(3) Navigation Support*.

**(1) Human Activity Monitoring.** We track and analyze human activity in real time as in Sec. 3. When collisions occur, the agent reverts to its prior position, and we identify whether the obstacle is human or an inanimate object. For human collisions, we log the coordinates and motion state to inform potential reward-shaping strategies. Distance and orientation estimates derive from agent–human coordinate data. For instance, we employ the Grounding-DINO Liu et al. (2024) detec-

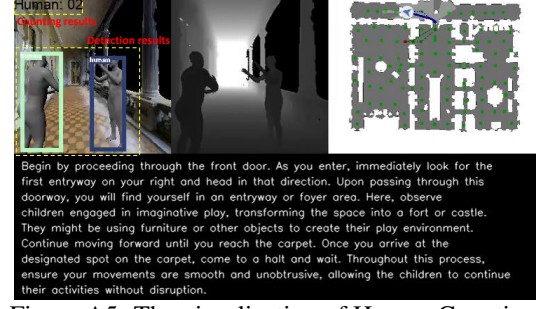

Figure A5: The visualization of Human Counting.

tor on RGB inputs with the prompt *"human"* to count individuals. Fig. A5 illustrates how human detection bounding boxes enable real-time counting.

**(2) Environmental Perception.** We maintain a dynamic scene graph comprising static elements (e.g., buildings, furniture) and moving humans. The agent continuously updates this graph by fusing positional changes and human motion data in its vicinity. This ensures real-time awareness of human activities for downstream decisions.

**(3) Navigation Support.** An A*-based planner computes candidate trajectories while accounting for both dynamic humans and static obstacles. During execution, we monitor any divergence between the agent's chosen route and the planner's recommended path. This method highlights human-centric obstacles and informs the agent's short-term re-planning steps. Our unified API supports real-time detection, tracking, and socially compliant navigation decisions in both *discrete* and *continuous* modes. It simplifies multi-human scene management, ensures intuitive collision handling, and provides robust path-planning assistance—together forming a foundation for advanced human-aware navigation algorithms.

### B.8  HUMAN ACTIVITIES ANNOTATION DATA ANALYSIS

**Human Distribution by Region.** Fig. A4(a) illustrates the distribution of 910 humans across 26 region types in 90 buildings, averaging about nine individuals per building. Even though each person moves independently, this distribution ensures robust and dynamic multi-human interactions, closely mirroring real-world scenarios.

**Motion Frequency Analysis.** Figs. A4(b)–(c) compare the 15 most frequent motions before and after multi-human enrichments. While the total number of motions increases, we also embed additional movement patterns and group interactions into existing actions. For instance, "*talking on the phone while pacing*" may now involve extended pacing distances or layered scenarios like "*a couple having a quiet conversation*" or "*friends sharing stories over dinner.*"

**Movement Distance Analysis.** Fig. A4(d) displays the distribution of trajectory lengths for actively moving humans. Specifically, 22.4% cover distances up to 0.5 m, 37.3% reach 0.5–1 m, 25.0% span 1–1.5 m, 11.6% extend 1.5–2 m, and the remaining 3.7% exceed 2 m. This wide range reflects the diverse indoor and outdoor behaviors encompassed in the dataset.

**Human Impact Analysis.** As shown in Fig. A4(e), humans exert a notable influence on navigation paths: 35.5% of the 16,844 paths in HA-VLN physically intersect with human motion, while 49.7% of viewpoints are indirectly affected (i.e., humans are visible along the route). These statistics underline the importance of accounting for human presence and movement trajectories when designing real-world navigation agents.

## C  AGENT DETAILS

### C.1  HA-R2R INSTRUCTION EXAMPLES

Table A3 illustrates four sample instructions from the *Human-Aware Room-to-Room* (HA-R2R) dataset. These examples encompass multiple scenarios: multi-human interactions (e.g., 1, 2, 3), direct agent–human encounters (e.g., 1, 2, 3), situations with four or more bystanders (e.g., 3), and paths devoid of humans (e.g., 4). Together, they demonstrate how HA-R2R challenges an agent with diverse human-aligned instructions.

### C.2  HA-R2R INSTRUCTION GENERATION

To create enriched instructions for HA-R2R, we use ChatGPT-4o and LLaMA-3-8B-Instruct to expand upon R2R-CE's original textual data. Our strategy involves a carefully crafted few-shot prompt, combining a *system prompt* (Listing 3) and *few-shot examples* (Listing 4).

**Prompt Structure.** The system prompt lays out guidelines for generating instructions that emphasize social context. It encourages mentioning human activities and interactions relevant to navigation paths Wu et al. (2025). Few-shot examples then illustrate the desired format, including references

Table A3: **Instruction Samples from the HA-R2R Dataset.** Text in purple highlights *human-related actions/movements*, while text in blue indicates explicit *agent-human interaction* cues. These examples illustrate how HA-R2R integrates dynamic human considerations and social awareness into navigation instructions.

| |
|---|
| 1. Exit the library and turn left. As you proceed straight ahead, you will enter the bedroom, where you can observe a person actively searching for a lost item, perhaps checking under the bed or inside drawers. Continue moving forward, ensuring you do not disturb his search. As you pass by, you might see a family engaged in a casual conversation on the porch or terrace, be careful not to bump into them. Maintain your course until you reach the closet. Stop just outside the closet and await further instructions. |
| 2. Begin your path on the left side of the dining room, where a group of friends is gathered around a table, enjoying dinner and exchanging stories with laughter. As you move across this area, be cautious not to disturb their gathering. The dining room features a large table and chairs. Proceed through the doorway that leads out of the dining room. Upon entering the hallway, continue straight and then make a left turn. As you walk down this corridor, you might notice framed pictures along the walls. The sound of laughter and conversation from the dining room may still be audible as you move further away. Continue down the hallway until you reach the entrance of the office. Here, you will observe a person engaged in taking photographs, likely focusing on capturing the view from a window or an interesting aspect of the room. Stop at this point, ensuring you are positioned at the entrance without obstructing the photographer's activity. |
| 3. Starting in the living room, you can observe an individual practicing dance moves, possibly trying out new steps. As you proceed straight ahead, you will pass by couches where a couple is engaged in a quiet, intimate conversation, speaking softly to maintain their privacy. Continue moving forward, ensuring you navigate around any furniture or obstacles in your path. As you transition into the hallway, notice another couple enjoying a date night at the bar, perhaps sharing drinks and laughter. Maintain a steady course without disturbing them, keeping to the right side of the hallway. Upon reaching the end of your path, you will find yourself back in the living room. Here, a person is checking their appearance in a hallway mirror, possibly adjusting their attire or hair. Stop by the right candle mounted on the wall, ensuring you are positioned without blocking any pathways. |
| 4. Begin by leaving the room and turning to your right. Proceed down the hallway, be careful of any human activity or objects along the way. As you continue, look for the first doorway on your right. Enter through this doorway and advance towards the shelves. Once you reach the vicinity of the shelves, come to a halt and wait there. During this movement, avoid any obstacles or disruptions in the environment. |

to human behavior (e.g., "*someone quietly making a phone call; keep your voice down as you proceed*"), positional references, and object interactions.

**Iterative Refinement.** In early trials, the models sometimes produced extraneous or subjective content, lacking sufficient detail on human activities. We iteratively refined the system prompt and examples, clarifying the need for neutral tone, accuracy, and contextual alignment with human-related scenarios. In each round, we analyzed model outputs, identified discrepancies, and adjusted examples to showcase more detailed, coherent, and socially aware instructions. This process guided ChatGPT-4o and LLaMA-3-8B-Instruct toward generating instructions that fully integrate human-centric elements—such as bystander activities, relevant spatial cues, and subtle behavioral recommendations. The final HA-R2R instructions thus reflect enriched scene descriptions where agents must account for diverse, real-world nuances involving human presence.

## C.3 HA-R2R DATA ANALYSIS

**Word Frequency Analysis.** We conduct a word frequency study on HA-R2R to gauge its capacity for representing realistic, human-centric scenarios. Figs. A6(a) and (b) illustrate frequently used nouns and verbs, confirming the dataset's focus on both spatial navigation and social interactions.

*Nouns.* The five most common nouns are *room*, *hallway*, *turn*, *area*, and *path*, with *room* alone appearing over 15,000 times. Other notable terms (*person*, *doorway*, *kitchen*) highlight spatial complexity and social elements such as *conversation*, *activities*, and *someone*.

*Verbs.* The five most frequent verbs—*is*, *continue*, *proceed*, *ensuring*, *be*—reveal an action-oriented narrative, while additional terms (*engaged*, *observe*, *notice*, *avoid*, *maintain*) underscore instructions geared toward social awareness and precise route-following.

```
1296
1297   "system": "You are tasked with
1298       generating an objective and
1299       detailed description of a path and
           the related scene information based
1300        on the given input. Focus solely
1301       on observable entities and actions
1302       in the scene, avoiding descriptions
1303        of smell, atmosphere, or
1304       subjective feelings. Ensure clarity
            and precision in object and human
1305       activity descriptions."
1306
1307   "examples": ...   //examples in Listing
1308       4
1309
       "user": Input:
1310   [instruction_text]: \"...\";
1311   [start]: \"...\";
1312   [during]: \"...\";
1313   [end]: \"...\";
1314
       Please provide an objective, step-by-
1315       step description of the path,
1316       focusing on the objects and human
1317       activities observed at each stage
           of the navigation (beginning,
1318       during, and end). Expand your
1319       description by including character
1320       actions, objective descriptions of
1321       objects, and details of the
1322       movement process. Your description
           should maintain a neutral tone and
1323       avoid embellishments. Please
1324       simulate the robot's movement along
1325        the path, while incorporating
1326       possible connections or
           interactions between the robot,
1327       objects, and human activities.
1328
1329   Output: ...   //generation
1330
```

Listing 3: Format of our LLM prompt for HA-R2R instrcution generation.

```
# Example 1:
Input:
[instruction_text]: \"Go
    through the doorway to the
    hall.  Go down the hall
    past the landing.  Take the
     first door on the right
    and stop in the doorway.
    \";
[start]: \"other room:A person
    using the room for a quiet
    phone call. \";
[during]: \"other room:A person
     using the room for a quiet
     phone call. \";
[end]: \"hallway:An individual
    cleaning or vacuuming. \";

Output:
You should start from the room.
     Before you walk through
    the doorway into the
    hallway, you will notice
    someone quietly making a
    phone call, so please
    remain quiet as you move.
    Then, proceed down the
    hallway past the landing.
    As you continue through the
     corridor, you will see a
    responsible cleaner using a
     vacuum to tidy up. Finally
    , your destination is at
    the end of the hallway,
    enter the first door on the
     right and stop in the
    doorway.
```

Listing 4: Few-shot examples for HA-R2R instrcution generation.

**Human Impact Analysis.** Fig. A6(c) shows that most instructions contain 20–60% human-related content, reflecting the dataset's emphasis on people in everyday scenes.

Comparisons of word clouds in Figs. A6(d) and (e) confirm that while both human-aligned and non-human segments use common navigational verbs (*walk, left, right*), instructions involving humans introduce additional social context (*couple, man, painting*). This integration of interpersonal cues elevates HA-R2R beyond simple route directives, better mirroring real-world navigation challenges in human-filled environments.

### C.4    VISUAL AND DEPTH EMBEDDINGS

Following VLN-CE Krantz et al. (2020), we employ parallel streams to process RGB and depth images. Each viewpoint produces a set of features from two specialized ResNet-50 models:

1. **RGB Features.** Let $\{v_1^{rgb}, v_2^{rgb}, \ldots, v_k^{rgb}\}$, where $v_i^{rgb} \in \mathbb{R}^{2048}$, be outputs of a ResNet-50 pretrained on ImageNet.

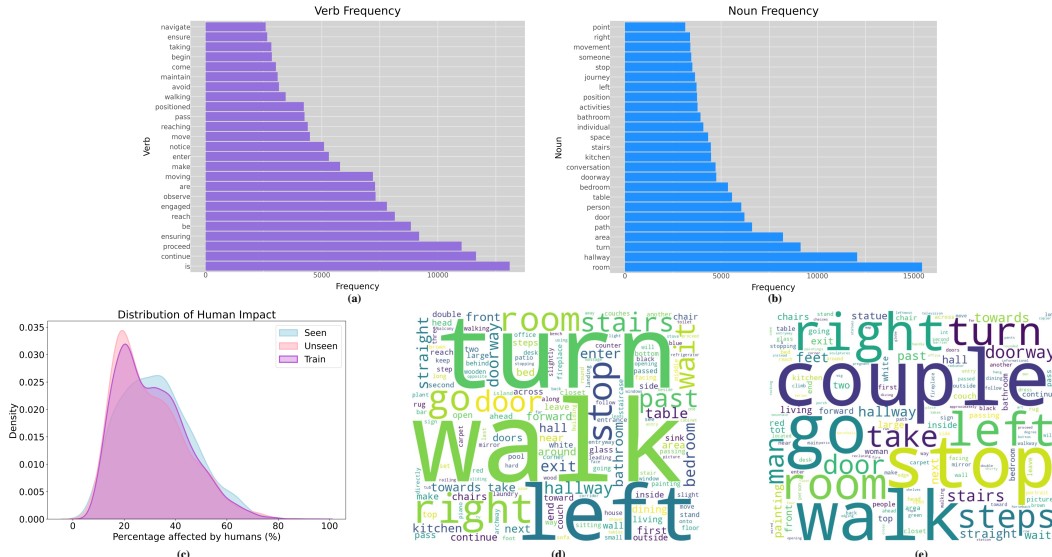

Figure A6: **Statistics for the HA-R2R Dataset.** (a) Verb frequency distribution for all instructions. (b) Noun frequency distribution for all instructions. (c) Distribution of human impact within HA-R2R (originally Fig. A4 in the main text; figure numbering differs due to inserted figures). (d) Word cloud of instructions not aligned with human activities. (e) Word cloud of instructions explicitly involving human actions. Larger font size indicates higher frequency or proportion in the dataset.

2. **Depth Features.** Let $\{v_1^d, v_2^d, \ldots, v_k^d\}$, where $v_i^d \in \mathbb{R}^{128}$, be outputs of another ResNet-50 pre-trained on Gibson-4+ Xia et al. (2018) and MP3D for point-goal navigation.

We fuse these two feature streams along with a directional encoding $d_i$ indicating spatial orientation:

$$v_i = \left[\, v_i^{rgb}\, W_{\text{rgb}};\ v_i^d\, W_{\text{depth}};\ d_i \,\right] W_{\text{merge}}, \tag{A2}$$

where $W_{\text{rgb}}$, $W_{\text{depth}}$, and $W_{\text{merge}}$ are learnable projection matrices with ReLU activation. The directional encoding $d_i$ is constructed by repeating $(\cos\theta_t^i,\ \sin\theta_t^i)$ 32 times, where $\theta_t^i$ measures the relative heading offset of the agent. The fused embedding $v_i \in \mathbb{R}^d$ is either 512 or 768 dimensions, matching the requirements of our **HA-VLN-CMA** or **HA-VLN-VL** agent, respectively. Both ResNet backbones remain fixed during training, ensuring consistent and stable representations from the RGB and depth channels throughout the learning process.

C.5    TEXT EMBEDDINGS

For the **HA-VLN-VL** agent, we utilize text embeddings from *PREVALENT* Hao et al. (2020), which was pre-trained on 6.58M image–text–action triplets, thereby capturing broad contextual cues for navigation. Conversely, the **HA-VLN-CMA** agent adopts embeddings from *BERT* Devlin (2018), also widely used for its strong language representations.

Formally, let $\ell = \{w_1, \ldots, w_n\}$ be a sequence of tokens representing the instruction. Each token $w_i$ is mapped to a one-hot vector $e_i \in \mathbb{R}^V$, where $V$ is the vocabulary size. An embedding matrix $E \in \mathbb{R}^{V \times d}$ then projects $e_i$ into a continuous $d$-dimensional space:

$$x_i = E^\top e_i, \quad x_i \in \mathbb{R}^d. \tag{A3}$$

In this manner, each discrete token $w_i$ is transformed into a trainable embedding $x_i$, forming the foundation of the model's linguistic understanding.

C.6    HA-VLN-VL STRUCTURE

**Model Overview.** **HA-VLN-VL** adopts a BERT-like architecture inspired by Recurrent VLN-BERT Hong et al. (2021), extending it to handle human-aware navigation. At each timestep $t$, the model receives the previous state $s_{t-1}$, language tokens $X$, and fused RGB–depth visual features $V_t$

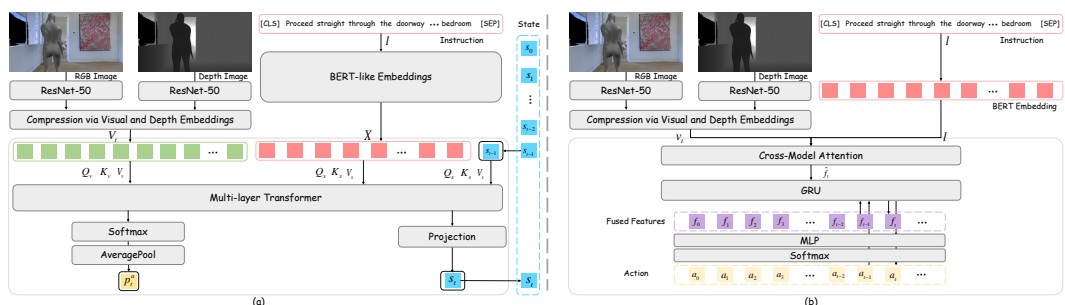

Figure A7: **Network Structures.** (a) **HA-VLN-VL** adopts a BERT-like transformer with a specialized state token. RGB and depth inputs are compressed by ResNet-50 and concatenated, while instruction tokens feed a BERT-like encoder. A multi-layer transformer computes cross-modal attention, producing per-step action probabilities via average-pooling and a final projection. In both architectures, continuous or discrete commands are then derived for navigation based on the agent's policy output. (b) **HA-VLN-CMA** employs a cross-modal attention (CMA) module combined with a GRU policy. RGB and depth images are first processed by two ResNet-50 encoders and fused into a single feature stream, which attends to the instruction tokens; the fused features are then fed into a GRU and MLP to predict actions.

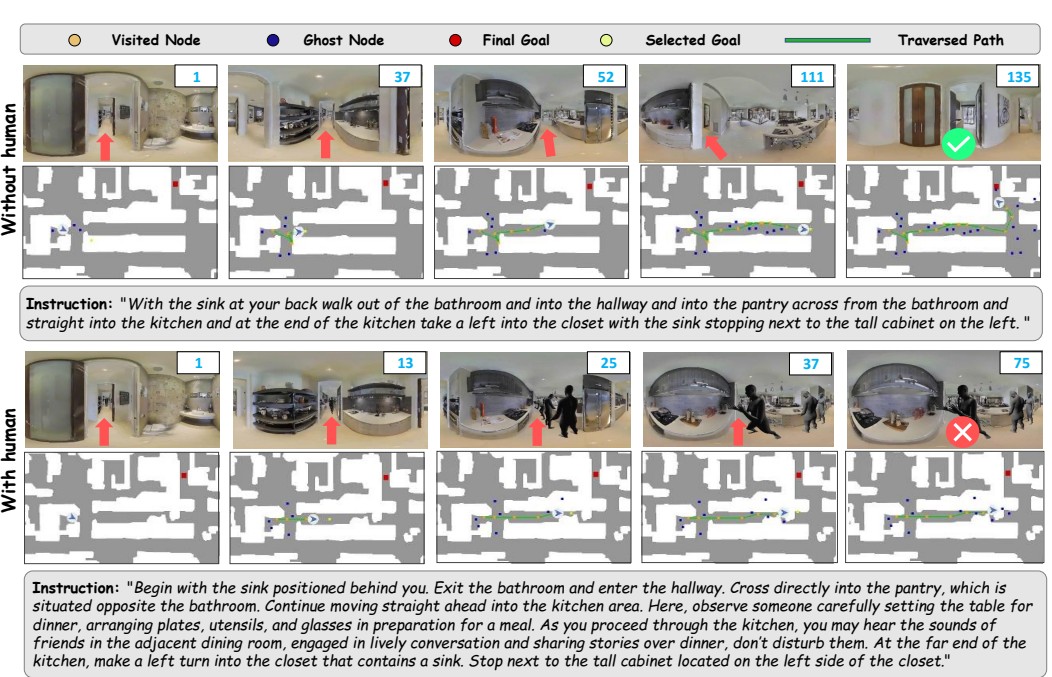

Figure A8: **Trajectory Comparison Under Human vs. No-Human Conditions.** We illustrate the same episode's trajectories predicted by BEVBert An et al. (2023), trained on VLN-CE, in scenarios with (bottom) and without (top) human presence. In the top row, no bystanders are present, and the agent follows its instructions with minimal collision risk. In the bottom row, bystanders and human-aligned cues lead to altered motion decisions, sometimes creating additional collision challenges or deviations.

(Sec. C.4). It outputs an updated state $s_t$ and an action distribution $p_t^a$:

$$s_t,\ p_t^a = \text{HA-VLN-VL}\big(s_{t-1},\ X,\ V_t\big). \tag{A4}$$

**State Token.** In line with BERT conventions, the model maintains a *state token* $s_t$ that encapsulates the agent's internal context. Initially, $s_0$ is set to embedding of `[CLS]` token. At each step, the state token is updated by appending agent's previously executed action $a_t$ and projecting resulting vector:

$$s_t = [\, s_t';\ a_t \,]\, W_s, \tag{A5}$$

where $s_t'$ is the final Transformer-layer output, and $W_s$ is a learnable projection matrix.

**Visual Attention.** To decide the next action, we compute attention scores between $s_t$ and the set of visual tokens $V_t = \{v_1, v_2, \ldots, v_n\}$, covering navigable directions plus a "stop" option:

$$A_{s,v}^t = \text{Softmax}\Big(\frac{Q_s\, K_v^\top}{\sqrt{d_h}}\Big), \tag{A6}$$

where $Q_s$ is derived from $s_t$ and $K_v$ from $v_i \in V_t$. The model then aggregates these attention scores via an average-pooling step:

$$p_t^a = \overline{\text{AveragePool}}\big(A_{s,v'}^t\big), \tag{A7}$$

yielding an action distribution over possible moves. The agent selects:

$$a_t = \arg\max\big(p_t^a\big). \tag{A8}$$

**Training Objective.** **HA-VLN-VL** is optimized through a combination of *supervised imitation learning*—to mimic ground-truth trajectories—and optional *reinforcement learning*, which rewards safe and efficient paths. As depicted in Fig. A7(a), the model continuously refines its understanding of language instructions and visual cues, offering robust and socially aware navigation.

## C.7 HA-VLN-CMA STRUCTURE

**Architecture Overview.** **HA-VLN-CMA** is a dual-stream visual-language agent featuring *Cross-Modal Attention (CMA)* and a recurrent decoder for navigation in human-populated scenarios (see Fig. A7(b)). It processes two visual channels—RGB and Depth—alongside language instructions, then outputs an action at each time step.

**Dual-Stream Visual Encoding.** Following Sec. C.4, each observation $o_t$ is split into:

$$v_t^{\text{rgb}} = \text{ResNet}^{\text{rgb}}(o_t), \quad v_t^{\text{d}} = \text{ResNet}^{\text{depth}}(o_t), \tag{A9}$$

where $\text{ResNet}^{\text{rgb}}$ and $\text{ResNet}^{\text{depth}}$ are separate backbones for RGB and Depth, respectively. The fused feature representation is

$$v_i = \big[v_i^{\text{rgb}} W_{\text{rgb}};\ v_i^d W_{\text{depth}};\ d_i\big]\, W_{\text{merge}}, \tag{A10}$$

where $W_{\text{rgb}}$, $W_{\text{depth}}$, and $W_{\text{merge}}$ are projection matrices, and $d_i$ is a direction encoding (Sec. C.4).

**Language Encoder.** Textual instructions $\{w_1, \ldots, w_T\}$ are transformed into contextual embeddings

$$l = \text{BERT}(w_1, \ldots, w_T). \tag{A11}$$

These embeddings capture the semantic structure of the instruction and serve as input to the cross-modal module.

**Cross-Modal Attention & Recurrent Decoding.** At time step $t$, we attend to the language features using multi-head attention:

$$\hat{f}_t = \text{MultiHeadAttn}(v_t,\ l), \tag{A12}$$

where $\text{Attention}(Q, K, V) = \text{softmax}\big(\frac{QK^\top}{\sqrt{d_k}}\big)V$. Multi-head attention helps handle lengthy and detailed instructions by learning multiple representations in parallel.

Next, we combine the resulting multimodal embeddings with the previous action $a_{t-1}$ in a GRU-based decoder:

$$f_t = \text{GRU}\big(\big[\,(v_t, l),\ a_{t-1}\,\big],\ f_{t-1}\big), \tag{A13}$$

where $f_{t-1}$ is the previous hidden state. Finally, an MLP outputs the action distribution:

$$a_t = \mathrm{softmax}(\mathrm{MLP}(f_t)), \tag{A14}$$

where $\mathrm{MLP}(f_t) = W_a\, f_t + b_a$, and $a_t$ is sampled from $P(a_t|f_t)$.

**Training Objectives. HA-VLN-CMA** is trained end-to-end with a mixture of imitation learning (to mimic ground-truth paths) and reinforcement learning (to encourage collision-free, socially compliant navigation). By learning from both paradigms, the agent refines its ability to balance path efficiency and safe distancing in human-populated environments.

# D  EXPERIMENTS DETAILS

## D.1  EVALUATION METRICS

We adopt a two-tier evaluation protocol for *HA-VLN*, measuring both *perception* (human awareness) and *navigation* (task completion). Perception metrics track how effectively the agent detects and responds to dynamic humans, while navigation metrics assess overall performance.

**Total Collision Rate (TCR).** Given the strong impact of human activities around critical nodes (viewpoints), we manage dynamic humans to ensure precise measurement. For navigation instance $i$, let $A_i^c$ be the set of human activities at these critical nodes. We define:

$$\mathrm{TCR} = \frac{\sum_{i=1}^{L}(c_i - |A_i^c|)}{L}, \tag{A15}$$

where $c_i$ counts collisions within $1\,\mathrm{m}$ of a human. TCR quantifies how often collisions occur in human-occupied zones.

**Collision Rate (CR).** CR is the fraction of navigation instances incurring at least one collision, conditioned on the fraction $\beta$ of instructions influenced by humans:

$$\mathrm{CR} = \frac{\sum_{i=1}^{L} \min(c_i - |A_i^c|, 1)}{\beta L}. \tag{A16}$$

Unlike TCR, CR highlights whether a collision occurred at all—offering insight into safety over entire trajectories.

**Navigation Error (NE).** NE is the mean distance between agent's final position and intended target:

$$\mathrm{NE} = \frac{\sum_{i=1}^{L} d_i}{L}, \tag{A17}$$

where $d_i$ is the agent–target distance at episode end.

**Success Rate (SR).** SR measures the ratio of episodes completed with zero collisions, and checks if the agent stops sufficiently close to the goal Anderson et al. (2018), we provide the equation for the collision check part here:

$$\mathrm{SR} = \frac{\sum_{i=1}^{L} \mathbb{I}(c_i - |A_i^c| = 0)}{L}, \tag{A18}$$

where $\mathbb{I}$ is 1 if the agent avoids collisions, and 0 otherwise.

## D.2  GROUND TRUTH PATH ANNOTATION

In HA-VLN-CE, the agent must reach within $3\,\mathrm{m}$ of the target while minimizing collisions. To label ground-truth paths, we use an A*-based heuristic search that identifies the shortest viable route, dynamically re-planning when obstacles block progress.

## D.3  FURTHER DISCUSSION ON STEP SIZE

In Table A4, a $1.0\,\mathrm{m}$ step was treated as four $0.25\,\mathrm{m}$ sub-steps, and a $2.25\,\mathrm{m}$ step as nine $0.25\,\mathrm{m}$ sub-steps, with collisions checked after each sub-step. When evaluated on the val_unseen split, BEVBert agent fails to navigate effectively with both $1.0\,\mathrm{m}$ and $2.25\,\mathrm{m}$ step sizes (SR drops to zero).

Table A4: **Impact of Step Size Combination on Navigation.** In this experiment, we treat 1m step as four 0.25m steps, and 2.25m step as nine 0.25m steps. In this case, collisions are detected every 0.25m. We show results for **BEVBert** An et al. (2023) on unseen validation.

| Step Size | NE↓ | TCR↓ | CR↓ | SR↑ |
|---|---|---|---|---|
| 1.00 | 6.85 | 26.97 | 0.94 | 0.004 |
| 2.25 | 8.79 | 112.78 | 0.97 | 0.000 |

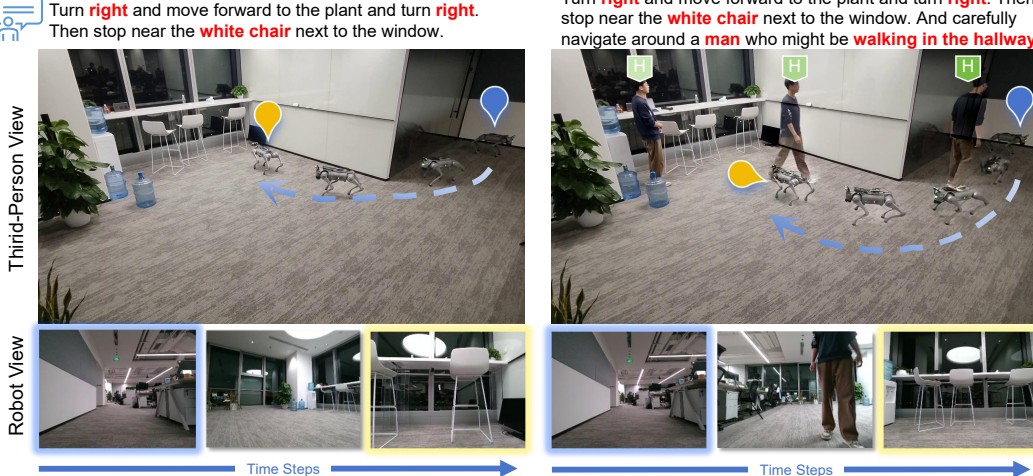

Figure A9: **Navigation success in an office** (*left*: no humans, *right*: with humans). *Top*: The given instruction for the robot. *Middle*: A third-person view of the robot's path. *Bottom*: The robot's selected view.

Table A5: **Navigation success rate across different region layouts** with (w/) and without (w/o) human presence. Each result is averaged over 30 episodes across 3 instances of each region type.

| | Living Room | | Office | | Hallway | | Lobby | | ALL | |
|---|---|---|---|---|---|---|---|---|---|---|
| Methods | w/o | w/ | w/o | w/ | w/o | w/ | w/o | w/ | w/o | w/ |
| HA-VLN-CMA-Base (trained on VLN-CE) | 0.23 | 0.08 | 0.26 | 0.08 | 0.30 | 0.13 | 0.24 | 0.07 | 0.26 | 0.09 |
| HA-VLN-VL (trained on VLN-CE) | 0.38 | 0.11 | 0.38 | 0.10 | 0.47 | 0.17 | 0.38 | 0.10 | 0.40 | 0.12 |
| HA-VLN-CMA-Base (trained on HA-VLN) | 0.24 | 0.13 | 0.24 | 0.13 | 0.29 | 0.20 | 0.23 | 0.13 | 0.25 | 0.15 |
| HA-VLN-VL (trained on HA-VLN) | 0.42 | 0.17 | 0.43 | 0.17 | **0.49** | **0.20** | 0.43 | 0.17 | 0.44 | 0.18 |

## D.4 VISUALIZATION OF NAVIGATION

Figs. A8 & 5 illustrate trajectories predicted by **BEVBert** An et al. (2023) (trained on VLN-CE) and **HA-VLN-CMA**∗, which showcases success and failure in human-filled or empty environments.

**Failures with Human Crossing.** In Fig. A8, the agent performs well when no bystanders are present. Yet in a human-populated setting, it fails to adjust at step 37 when a volunteer crosses its path, leading to collision.

**Collision vs. Avoidance.** Fig. 5 similarly shows two scenarios. At step 39 in the top pane, a direct approach by a bystander overwhelms the agent, causing a collision. In the bottom pane at step 22, the agent successfully deviates upon sensing a person nearby, avoiding any collision altogether. These visualizations confirm that dynamic human presence greatly complicates navigation, highlighting the need for robust social-aware models.

## D.5 VALIDATION ON REAL-WORLD ROBOTS

To deploy our navigation agents on physical hardware, the robot is equipped with an *NVIDIA Jetson NX* for AI inference and a *Raspberry Pi 4B* for motion control. The Jetson handles core navigation computations (receiving camera images and inferring action commands), while the Pi executes high-level movement directives such as *turn left* or *move forward*. We set a minimum step size of

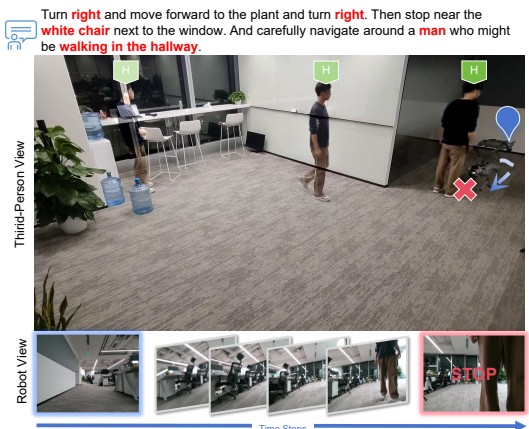

Figure A10: **Navigation failure in an office setting.** A volunteer abruptly changes position, causing robot to collide mid-path. This highlights the difficulty of adapting to sudden human movement in confined workspaces.

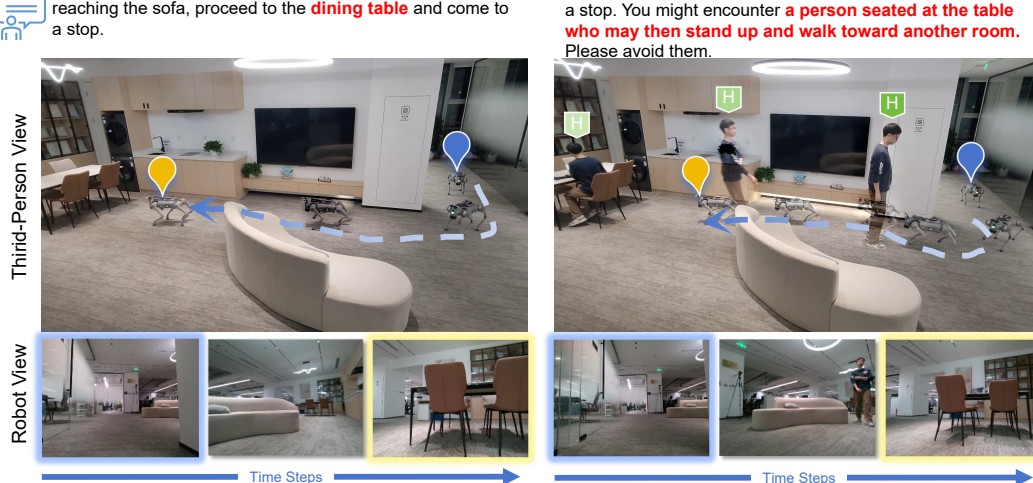

Figure A11: **Navigation success in a living room** (*left*: no bystanders, *right*: with bystanders). The robot follows instructions toward the sofa and dining area, keeping safe distances while navigating around volunteers.

0.25 m and a rotation increment of 15 degrees. An onboard IMU continuously monitors the robot's orientation and position, ensuring movement commands align with issued directives.

**Setup.** Our evaluations use a *Unitree GO2-EDU* quadruped, featuring the *Intel Realsense D435i* camera providing RGB imagery and a *3D LiDAR* below camera for detection. IMU refines positional and orientational control, enabling consistent motions. The quadruped rotates to get the panoramic view at each step. We evaluate our agents in four types of everyday indoor environments (each with three instances)—*office*, *living room*, *hallway*, and *lobby*—under two conditions: *(i)* w/o human presence (no bystanders) and *(ii)* w/ human presence (2-4 free-moving volunteers). This setup simulates realistic indoor traffic patterns and partial observability.

**Observations.** As illustrated in Fig. 6 (b), the robot frequently pauses or yields to avoid oncoming pedestrians. In the absence of bystanders, it navigates smoothly (Fig. A9), but collisions arise in cramped corridors or when crowds converge suddenly (Fig. A10). We observe similar patterns in living-room environments (Figs. A11–A12) and hallways (Fig. A13).

Table A5 shows the average **NSR** (Navigation Success Rate) across 30 trials in each instance. While human presence invariably lowers **NSR**, HA-VLN-VL consistently outperforms HA-VLN-CMA-Base, demonstrating stronger adaptability to dynamic motion. Also, Table A5 shows agents trained

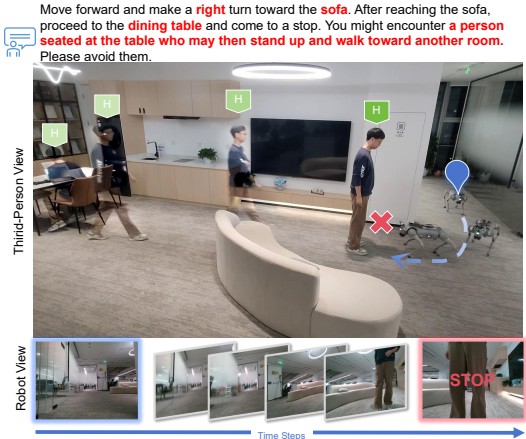

Figure A12: **Navigation failure in a living room with multiple bystanders.** Attempting to move beyond sofa toward a dining area, the robot collides with a volunteer who abruptly stands and shifts position. This underscores how unpredictable human motion can disrupt agent's intended path, requiring rapid re-planning.

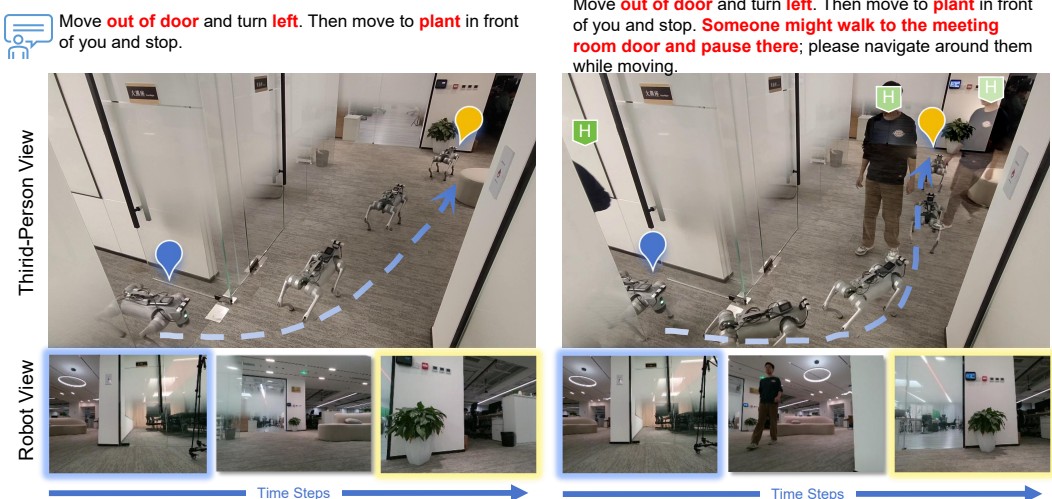

Figure A13: **Navigation success in a hallway** (*left*: no bystanders, *right*: with bystanders). When volunteers appear, the robot halts or deviates to avoid collisions, showcasing adaptive behavior in a constrained corridor.

on HA-VLN achieve higher **NSR** (0.18 vs. 0.12) than VLN-CE, demonstrating HA-R2R's sim-to-real gain under realistic conditions. Still, partial observability and abrupt group formations remain challenging, especially in narrow passages or at congested junctions. Appendix D.5 further details performance under varying crowd densities.

**Visual Demonstrations.** Figs. A9, A11, and A13 show the robot traversing distinct indoor environments—offices, living rooms, and hallways—guided by natural-language instructions. In Fig. 6 (b), the robot navigates around multiple people, leveraging camera inputs to avoid collisions through minor path adjustments. Although the agent typically succeeds in reaching its destination, collisions remain possible when bystanders change their trajectories unexpectedly. Figs. A10, A12, and A14 illustrate such scenarios, highlighting real-time challenges in unpredictable, human-inhabited spaces. More demos including a compilation video on our project webpage, further illustrate robot's performance and underscore how human-aware training aids sim-to-real transfer in dynamic indoor environments.

**Insights.** These experiments confirm that simulation-trained, multi-human navigation policies can indeed transfer to physical robots. However, further refinement in collision forecasting and reactive control is needed to handle unpredictable human behavior in tight indoor settings.

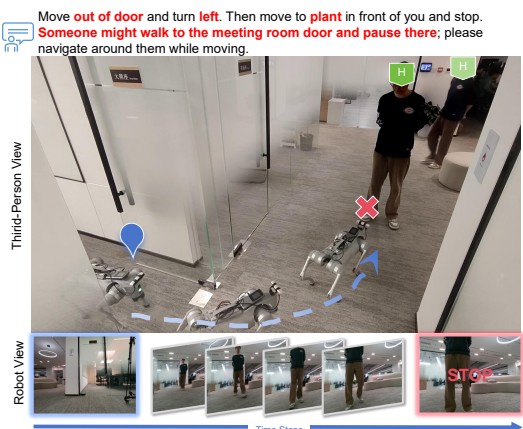

Figure A14: **Navigation failure in a hallway.** A volunteer's sudden positional change causes a mid-path collision and mission failure, reflecting the challenge of unpredictable human movement, even in comparatively open corridors.

# E USE OF LLMS

Large Language Models (LLMs) were used to aid in the writing and polishing of the manuscript. Specifically, we used an LLM to assist in refining the language, improving readability, and ensuring clarity in various sections of the paper. The model helped with tasks such as sentence rephrasing, grammar checking, and enhancing the overall flow of the text.

It is important to note that the LLM was not involved in the ideation, research methodology, or experimental design. All research concepts, ideas, and analyses were developed and conducted by the authors. The contributions of the LLM were solely focused on improving the linguistic quality of the paper, with no involvement in the scientific content or data analysis.

The authors take full responsibility for the content of the manuscript, including any text generated or polished by the LLM. We have ensured that the LLM-generated text adheres to ethical guidelines and does not contribute to plagiarism or scientific misconduct.

