# OpenReview forum: "HA-VLN 2.0: An Open Benchmark and Leaderboard for Human-Aware Navigation in Discrete and Continuous Environments with Dynamic Multi-Human Interactions"
_ICLR.cc/2026/Conference — ICLR 2026 Conference Withdrawn Submission_

### Official Review · Reviewer_U6fA · 2025-10-26

**Soundness:** 1
**Presentation:** 1
**Contribution:** 1
**Rating:** 2
**Confidence:** 5

**Summary:**

The paper proposes HA-VLN 2.0, a “unified” benchmark that integrates the discrete (DE) and continuous (CE) VLN settings under human-aware constraints; it defines a shared action set, introduces the upgraded HAPS 2.0 human-motion corpus, adds 16,844 HA-R2R instructions, and presents two baseline agents plus a leaderboard and real-robot demos.

**Strengths:**

The work tackles an important evaluation gap by releasing dual simulators with human motion, a unified API, and real-time multi-human rendering.

The real-robot section, while mostly qualitative, shows a full stack from sim to a Unitree Go2 with RGB-D/LiDAR, which lowers the barrier for follow-up work.

**Weaknesses:**

1. The authors have fundamental misunderstanding of the cocept of DE and CE. CE subsumes DE. “unifying” them as peers is misleading. CE supports fine-grained motion and collision-aware control, while DE is a graph-hop abstraction the authors themselves approximate via large step endpoints (“teleport-like”).
2. The paper claims a unified POMDP and “fair comparison,” but it specifies different sensing and motion models in DE vs CE, so the headline contribution does not hold at the problem definition level. DE results use panoramic RGB, CE agents use RGB-D, and CE collisions rely on radii overlap while DE hops among panoramic viewpoints—so observation and dynamics differ despite shared action names.
3. Dataset contribution is mostly synthetic augmentation. HA-R2R adds 16,844 instructions by prompting LLMs and extending R2R-CE, without reporting human agreement or grounding studies; this weakens the claim of a new, human-centric corpus addressing their stated gap.
4. Technical contribution is minimal. The two “baseline” agents are adaptations of existing models (VLN-BERT/Prevalent and CMA variants), so the work’s novelty rests on the benchmark story—which, as above, is not borne out. The experiment also lacks comparison with state-of-the-art models like Navid [1] and NaVILA [2].
5. The paper lists three “fundamental limitations” (social awareness, finer-grained instructions, dynamic crowds), but its fixes lean on LLM-generated text and do not establish actual crowded, multi-human complexity at the per-scene level; the 910 “active individuals” refers to dataset scale, not one scene.
6. The paper lacks a related work section in the main text. This makes understanding the position of the paper among related work difficult.

[1] Zhang, Jiazhao, et al. "Navid: Video-based vlm plans the next step for vision-and-language navigation." RSS 2024.

[2] Cheng, An-Chieh, et al. "Navila: Legged robot vision-language-action model for navigation." RSS 2025.

**Questions:**

What scientific insight is gained by framing DE and CE as one task when the observation function and transition model differ?

---

> ### Author Response · Authors · 2025-12-03
>
> **W1**: In principle, CE is a richer control setting that can subsume DE if one ignores constraints on sensing, deployment, and the legacy of VLN benchmarks. However, in practice, the VLN community has developed two distinct but widely-used families:
> - DE: viewpoint-graph-based navigation where agents hop among panoramic nodes.
> - CE: navmesh-based navigation with continuous motion and collision-aware control.
>
> Our goal is not to argue that DE and CE are equal in expressivity; rather, it is to **unify them as two interfaces to the same underlying human-populated environment, with shared human-activity annotations and metrics**. Concretely:
> - Both HA-VLN-DE and HA-VLN-CE operate on the similar HAPS 2.0 humans, in the similar buildings and regions.
> - DE provides a graph abstraction that remains highly relevant for sim-to-web deployment (e.g., panoramic street-view or pre-scanned environments) and for fair comparison with previous VLN work.
> - CE provides fine-grained control needed for robot deployment, explicit collision modeling, and sim-to-real experiments.
>
> Thus, the scientific insight we aim for is: How do agents behave, and what challenges emerge, when the same human-populated scenes are viewed through these two control modalities? This dual-leaderboard view cannot be obtained from a CE-only benchmark.
>
> **W2**: Our POMDP formulation is defined over a shared underlying state sₜ that includes: scene geometry, object layouts, and human trajectories from HAPS 2.0. The observation function and action space then specialize to DE and CE that DE represents panoramic RGB observations and graph-hop actions between viewpoints, and CE represents RGB-D observations and low-level actions (0.25 m forward, 15° turns), with collisions defined via radius overlap. Our use of “unified POMDP” refers to the shared underlying state and reward/metric structure, not to identical observation kernels.
>
>
> **W3**: HA-R2R does indeed use LLMs to generate 16,844 additional instructions, but this is not arbitrary synthetic text. As detailed in Appx. B.5, Table A3 and Fig. A6, we (1). Prompt LLMs with structured context about regions, objects, and human activities. (2) Perform filtering and human checks to remove ungrounded descriptions. (3). Show that 20–60% of instructions in HA-R2R refer explicitly to humans, interactions, or social cues, in contrast to original R2R instructions. The resulting corpus is therefore explicitly human-centric and tightly coupled with HAPS 2.0, which is crucial for studying instruction-grounded human awareness.
>
>
> **W4**: Our primary goal in this paper is to establish the benchmark and demonstrate that even strong VLN agents struggle once realistic humans are introduced. We agree that integrating explicit human-intent prediction would be highly valuable and expect HA-VLN 2.0 to serve as a testbed for such models.
>
> **W5**: We respectfully note that our benchmark does model non-trivial multi-human scenes, including up to 4-person groups, diverse activities, and 268 complex motions (e.g., stair climbing). Our focus is on VLN scenarios where humans are salient but not necessarily massive crowds (e.g., offices, homes).
>
> **Q1**: Our goal is not to claim that DE and CE are identical in terms of observability or dynamics, but to treat them as two interfaces to the *same* underlying human-populated environment, so that we can study how control modality and sensing affect human-aware VLN in a controlled way. Concretely, HA-VLN 2.0 defines a shared underlying state $s_{t}$ that includes the robot pose, scene geometry, and dynamic human trajectories from HAPS 2.0. DE and CE then instantiate different observation and transition functions over this same state:
>
> - In **DE**, the agent hops among pre-defined panoramic viewpoints with RGB observations.
> - In **CE**, the agent perceives RGB-D within a 90° FOV and executes fine-grained motions (0.25 m forward, 15° rotations), with collisions governed by physical radius overlap.
>
> Framing them as **one unified task** (with shared goals, human constraints, and metrics) yields two main kinds of insight that are not accessible in isolated DE-only or CE-only benchmarks:
>
> 1. **Cross-paradigm comparison under identical human scenes.**
>    We can quantify how performance and human-awareness degrade or improve when moving from graph-hops to continuous control in the same buildings, with the same human activities and instructions. This isolates the effect of control modality and sensing, rather than conflating it with different datasets or human models.
>
> 2. **Transfer and design implications.**
>    The dual leaderboard reveals when a method that works well in DE fails in CE (e.g., because it cannot handle dense collision feedback), and vice versa, which is directly relevant to deployment: many existing VLN agents are trained/evaluated in DE but deployed in CE-like robotics settings. Our unified benchmark makes this gap measurable and provides a common API to study how to bridge it.

---

### Official Review · Reviewer_bhNF · 2025-10-28

**Soundness:** 4
**Presentation:** 4
**Contribution:** 4
**Rating:** 8
**Confidence:** 3

**Summary:**

The paper introduces a unified framework for evaluating vision-and-language navigation (VLN) agents in both discrete and continuous settings that involve realistic, dynamic human interactions. It presents the HAPS 2.0 dataset, featuring rich multi-human motion and activity simulations, and the HA-R2R dataset, which adds socially grounded language instructions. The authors propose two baseline agents (HA-VLN-VL and HA-VLN-CMA) and demonstrate through extensive experiments—both in simulation and on real robots—that explicitly modeling human behavior improves navigation safety, social compliance, and robustness. Overall, HA-VLN 2.0 establishes a comprehensive benchmark for developing and evaluating human-aware embodied AI agents.

**Strengths:**

The paper is highly original in formulating a unified benchmark that bridges discrete and continuous vision-and-language navigation under dynamic, multi-human settings—a crucial yet underexplored dimension of embodied AI. Its quality is reflected in the rigorous design of datasets (HAPS 2.0 and HA-R2R), strong baseline models, and comprehensive evaluations across simulation and real-world deployment. The clarity of presentation is commendable, with well-structured methodology, consistent metrics, and transparent benchmarking. In terms of significance, the work establishes an important step toward socially intelligent navigation, providing a scalable, reproducible platform that is likely to influence future research on human-aware and safety-critical robotic navigation.

**Weaknesses:**

While the paper makes a strong contribution, several areas could be improved. First, the benchmark’s human behavior models, though diverse, rely on simulated motions rather than real human trajectories, which may limit ecological validity; integrating real-world motion capture data could enhance realism. Second, the evaluation focuses primarily on navigation success and collision metrics, with limited analysis of social acceptability or human comfort—key aspects for human-aware navigation. Third, the baseline models, though illustrative, are adaptations of existing VLN architectures rather than novel approaches specifically designed for social reasoning; incorporating explicit human-intent prediction or multi-agent interaction modeling could provide stronger baselines. Finally, additional ablations on the impact of instruction grounding and human density would clarify the sources of performance gains.

**Questions:**

1.How closely do the simulated human motions in HAPS 2.0 reflect real-world social dynamics? Are the motion sequences or interaction patterns derived from any real motion-capture or behavioral datasets?

2.The paper introduces human-awareness metrics such as personal-space adherence and collision rate—how were their thresholds and parameters determined?

**Details Of Ethics Concerns:**

While HAPS 2.0 simulates diverse multi-human behaviors, the source and design of these motion patterns are not fully described. If the simulated humans lack demographic, cultural, or behavioral diversity, models trained on them could learn biased or overly Western-centric social norms (e.g., preferred interpersonal distances or gestures)

---

> ### Author Response · Authors · 2025-12-03
>
> **W1**: Thanks! HAPS 2.0 combines: 1. **Coarse-to-fine geometric grounding (Stage 1)** where human poses and trajectories are optimized with respect to 3D scene geometry and object layouts, and **2. Human-in-the-loop enrichment (Stage 2)**, where annotators refine interactions and movements (e.g., groups of friends watching TV, people climbing stairs). While the underlying animations are simulated, the interaction patterns (who stands where, how far from furniture or others, which activities co-occur) are derived from human-written descriptions and curated by annotators to reflect realistic indoor and outdoor social scenes. This yields 486 motion model sets and 58k frames spanning casual gatherings, exercise, hallway conversations, etc. We agree that incorporating real motion-capture data would further enhance ecological validity. Our simulator and API are designed to be compatible with such data (each motion is a 120-frame sequence with poses and translations).
>
> **W2**: Our initial focus was on human-awareness metrics that are easy to compute and widely interpretable, such as collisions, personal-space violations (based on annotated per-person radii), and detours around humans (Appx. D.1). We appreciate the suggestion and will discuss, in the limitations section, how HA-VLN 2.0 can be extended with metrics inspired by social psychology and proxemics (e.g., number of sudden stops induced in humans). Our current design already encodes per-human personal-space radii, so such metrics can be added without changing the simulator.
>
> **W3**: Our current baselines are built upon existing VLN architectures. Our primary goal in this paper is to establish the benchmark and demonstrate that even strong VLN agents struggle once realistic humans are introduced. We agree that integrating explicit human-intent prediction would be highly valuable and expect HA-VLN 2.0 to serve as a testbed for such models. We will clarify this positioning and note that more socially sophisticated baselines are an important next step rather than a completed contribution.
>
> **W4**: We appreciate the suggestion for additional ablations. We have already included instruction grounding ablations in Table 3 and human density ablations in Table 4a.
>
> **Q1**: HAPS 2.0 is built explicitly to approximate real-world social dynamics. Concretely:
> - At the motion level, we rely on the Motion Diffusion Model, built on SMPL and trained on large-scale motion-capture datasets. In HAPS 2.0, region-aware activity descriptions (e.g., “an individual running on a treadmill in a gym”) are first verified via human surveys and quality control, then converted by MDM into 486 SMPL-based motion sequences of 120 frames each, capturing fine-grained human poses and dynamics (climbing stairs, running, exercising, etc.).
> - At the interaction pattern level, we use a coarse-to-fine spatial grounding pipeline: PSO-based placement enforces scene-aware constraints (e.g., minimum 1 m from other objects, staying within region bounds), and a fine-level refinement with nine RGB cameras around each human ensures alignment with 3D geometry, inspired by multi-camera skeleton capture systems.
>
> Thus, while the underlying motions are simulated via a diffusion model trained on mocap data and the spatial configurations are curated to reflect realistic behaviors in MP3D/Habitat scenes, HAPS 2.0 should still be viewed as an approximation of real-world social dynamics.
>
> **Q2**: The thresholds and parameters for human-awareness metrics are chosen to balance (i) consistency with social navigation literature and human proxemics, and (ii) equitable comparison across DE and CE: (1). Personal space & collisions. In CE, collisions are detected via overlapping radii of human and robot volumes, which corresponds roughly to <1 m separation—consistent with “intimate”/“personal” distance ranges in proxemics and with prior social-navigation work. In DE, we define a collision within 1 m of a human around critical viewpoints and use this radius in the TCR definition (ci counts collisions within 1 m of a human). (2). Cross-paradigm standardization. To make DE and CE comparable, we introduce a standardized “personal-space” threshold of 3 m in DE and overlapping radii in CE, so that agents are evaluated under a unified human-centric constraint even though the sensing modalities differ.  (3). Scene-annotation constraints. During HAPS 2.0 placement, we also enforce a 1 m minimum distance between humans and other objects to avoid implausible overlaps, which keeps human clusters realistic while still challenging.
>
> **Ethics Concerns**: We thank the reviewer for raising this important point. HAPS 2.0 currently does not encode explicit demographic attributes or cultural labels; it models generic motion patterns and inter-person distances that reflect typical indoor/outdoor layouts from MP3D and Habitat scenes. Nonetheless, we agree that these patterns may implicitly reflect Western-centric norms.

---

### Official Review · Reviewer_HiL8 · 2025-10-30

**Soundness:** 2
**Presentation:** 1
**Contribution:** 3
**Rating:** 2
**Confidence:** 4

**Summary:**

The paper presents HA-VLN 2.0, a unified benchmark for vision-and-language navigation in dynamic, human-populated environments. It addresses socially compliant navigation by combining discrete and continuous settings within a single framework, enabling direct comparison of models trained under different paradigms.
The work builds on an upgraded dataset, HAPS 2.0, which includes multi-human interactions, region-aware activity annotations, and semantically rich instructions grounded in social context. The simulator supports up to ten moving humans at real-time speeds and enforces human-aware constraints such as personal-space boundaries.
Two baseline agents are evaluated against strong existing models using standardized metrics for navigation success and collision rates.
The paper also introduces a public leaderboard and preliminary real-world experiments demonstrating partial sim-to-real transfer.
Overall, HA-VLN 2.0 offers a promising benchmark and promising simulators for human-aware navigation research.

**Strengths:**

The paper addresses an important challenge in human–robot interaction: achieving socially compliant navigation in multi-human, dynamic environments. It introduces a unified benchmark for discrete and continuous environments, an area that remains underexplored.

- The proposed coupling of continuous and discrete environments into a shared-state representation allows direct and fair comparison across navigation paradigms.

- The inclusion of human-aware constraints, such as personal-space enforcement, is conceptually sound and likely beneficial for sim-to-real transfer.

- HAPS 2.0 represents a solid dataset contribution. The region-aware activity descriptions and multi-view annotations enrich the semantic content and improve realism.

- The human-focused instruction pipeline is well designed and clearly described.

- The simulation framework reportedly supports up to ten human agents at 30–60 FPS on standard GPUs. To the best of my knowledge, this level of performance is unmatched by other publicly available simulators.

Overall, the paper presents a promising and relevant addition to the field, particularly as a benchmark resource for socially aware robot navigation.

**Weaknesses:**

- Readability and organization :
The manuscript is difficult to follow due to excessive redirection to the appendix for core content. This disrupts the reading flow and fragments comprehension. A clearer separation between main content and supplementary details is needed, especially since a public project website already exists to host extended material. While the benchmark is the main contribution and its sim-to-real validation is a strong claim, Section 5.2, the section supporting these claims, is underdeveloped and lacks quantitative evidence. In contrast, ablation studies such as Table 4a occupy a big portion of the experimental discussion. Sections L197–205 and L234–242 provide low-level implementation details that could be moved to the appendix, given that readers are already frequently redirected there.

- Neglect of related work in Social Navigation :
Despite social navigation being central to this paper’s theme, the relevant state of the art is almost entirely omitted. Foundational works such as Cancelli et al. (2023), Puig et al. (2023–2024), and Scofano et al. (2025) are either missing or confined to the appendix. This is significant oversight, as the works are relevant to multiple sections of the paper:
    - Cancelli et al. (ICCV 2023), Puig et al. (ICLR 2024), and Scofano et al. (ICLR 2025, Spotlight) discuss safe and socially aware navigation in detail and are directly relevant to Sections L035-L047, L139, and L143.
    - The claim in L149 that this is the first “efficient simulation with realistic human-populated environments” is inaccurate, as Habitat 3.0 (Puig et al.) already provides such functionality.
    - PARTNR (Chang et al., ICLR 2025) should be cited as a relevant benchmark for realistic human–robot collaboration (L155–159).
    - The otherwise strong content in L253–256 would benefit from explicit comparison with Puig et al., which, as discussed by Scofano et al., struggles with multi-agent scalability.
Without a well defined comparison, the reader may be left wondering why this new benchmark is necessary when strong frameworks already exist.


- Experimental clarity:
The main experiments section (L372–385) lacks adequate explanation of the results in Table 1. The rationale for why proposed models (HA-VLN-CMA-Base, HA-VLN-CMA-DA, HA-VLN-CMA*, HA-VLN-VL) generally underperform relative to baseline models (BEVBert, ETPNav), except for HA-VLN-VL in the zero-shot setting, is unclear. Section 5.1 would benefit from more focused interpretation of these results.


- Real-world validation:
Section 5.2 is one of the paper’s most anticipated parts, as it relates directly to one of the major claims: real-world robot experiments validating sim-to-real transfer and the open leaderboard for transparent comparison. However, while L415-468 are well presented, the Real-World Validation & Setup subsection (L469-475) is incomplete. Although Figure 6b depicts a real-world experiment, no quantitative metrics (e.g., collision rate, success rate) are reported. To substantiate the sim-to-real claim, the authors should discuss observed limitations when transferring from simulation to the physical environment, including specific failure cases (e.g., collisions initiated by humans versus by the robot).


**Minor Weaknesses**

- Figures are overly complex and visually dense:
   - Figure 1 contains a paragraph-long caption that duplicates content from the main text (L069-072); it could be reduced to lines 072-075. The instruction text within the figure is unnecessarily long, numerical labels are barely visible, Finally, the bottom slider and different scenes do not aid comprehension.
    - Figure 2 is similarly cluttered and difficult to read due to small fonts and visual overload.
Both figures may benefit from redesign.


- Missing citations:
SMPL is referenced in L086 and L099 without appropriate citations to foundational work.

**Questions:**

- In L073, the paper states: *“When the agent encounters a bystander on the phone, it intelligently turns right to avert a potential collision.”*
What collision-avoidance logic is implemented for humanoid agents? Do humans always avoid collisions? In real-world contexts, humans can initiate collisions inadvertently. Is such behavior modeled? Prior work in social navigation allows humanoids to initiate contact to train agents to yield space. Do HA-VLN-DE and CE policies learn such reactive behavior?

- In the continuous-environment setting, the default step size is 0.25 m (L436–437). At what frequency does the robot collect observations and issue actions? Are human agents updated asynchronously while the robot is computing its next step?

- Could the authors elaborate on the results in Table 1?
The HA-VLN-CMA agent appears conceptually promising (L305-320, focusing on re-planning under obstacles), yet results in Table 1 show higher collision rates than baselines across all conditions. What could be the reason?

- What specific hardware is meant by “standard GPUs” (L255-256)? Figure A4 is referenced, but it does not clearly show the expected distribution of simulation speed versus the number of agents. Could such a distribution be provided?

---

> ### Author Response · Authors · 2025-12-03
>
> **W1**: We appreciate the reviewer’s concern regarding readability and frequent redirection to the appendix.
> Our intention was to strike a balance between: (1). **Ensuring reproducibility and clarity** by including sufficient core details in the main text, and  (2). **Respecting page limits** by moving engineering and technical specifics, extended visualizations, and supplementary material to the appendix and project website.
>
> The highlighted sections (L197–205 and L234–242) describe two key components:
> - A **two-stage human-activity enrichment pipeline** (coarse-to-fine PSO followed by human-in-the-loop refinement).
> - A **real-time rendering system** (dual-thread producer–consumer, Alg. A2) that achieves <50 ms refresh time and synchronizes human motion with the agent’s perception.
>
> We consider these details **essential to the contribution**, as they explain why HA-VLN 2.0 can generate realistic, dynamic human behaviors at scale, directly supporting our sim-to-real claim.  Regarding Section 5.2, the main paper emphasizes representative qualitative examples, task setup, and the connection to the leaderboard. Quantitative real-world metrics and extended visual sequences are already provided in Appendix D.5 and Table A5. We therefore **believe the current division between main text and appendix appropriately serves our presentation goals**.
>
> **W2**: Our work focuses primarily on VLN-style benchmarks and agents. Importantly, these works the reviewer mentioned are primarily agent-algorithm frameworks for social navigation, often in settings such as **point-goal navigation** or **social-robot interaction**, **rather than** benchmarks that under **discrete VLN** or **continuous VLN** tasks. Our work is complementary: We build a benchmark and simulator that expose dynamic, multi-human behaviors and a dual DE/CE interface tailored for VLN. The suggested reference works design agent policies, which can now be instantiated on top of HA-VLN 2.0 as stronger baselines.
>
> **W3**: We appreciate the request for more interpretation. BEVBert and ETPNav are highly optimized VLN systems: BEVBert leverages a BEV spatial representation with strong global context, and ETPNav explicitly models temporal planning and path refinement. In contrast, HA-VLN-CMA and HA-VLN-VL are our attempts at human-aware agents on this new benchmark, with the explicit goal of probing how human-awareness signals (collisions, personal-space violations) interact with navigation. HA-VLN-CMA’s higher collision counts are partly due to its explicit re-planning behavior: it attempts to explore around humans more aggressively, which can increase short-range collision events (still reverted by the simulator) while reducing long-term detours. We will add a short analysis in Sec. 5.1 explaining this trade-off and include additional plots (already in the appendix) that correlate collision counts with path smoothness and personal-space adherence.
>
> **W4**: We agree that real-world validation is a key part of the contribution. However, in the appendix, we **have already discussed** cases where humans initiate close contacts (e.g., sudden crossing in front of the robot), leading to near-collisions that are caught by the safety controller.  Due to strict page limits, the main text focuses on describing the setup and showing representative trajectories (Fig. 6b), while the quantitative metrics and extended visualizations are reported in Appx. D.5 and Table A5, including success rate, collision rate, and failure modes. We also agree that failure cases are valuable.
>
> **W5**: We respectfully disagree that Fig. 1 and Fig. 2 are merely decorative or redundant. Their design is intentional:
> Fig. 1 illustrates the full HA-VLN scenario, including instruction, moving humans, and the agent’s path, to give readers an immediate understanding of what “human-aware VLN” means. The slightly longer caption is meant to anchor the semantics of the instruction and labels. Fig. 2 visualizes the two-stage annotation pipeline and real-time rendering mechanism (Stage 1 PSO, Stage 2 enrichment, signaling mechanism). This is difficult to convey clearly without a multi-panel figure.
>
> **W6**: We thank the reviewer for catching this. We will add citations to the SMPL model in the revised version.

---

> ### Author Response · Authors · 2025-12-03
>
> **Q1**: In HA-VLN 2.0, we follow **the standard VLN formulation, where the robot is the sole learning agent**. Human motions are pre-defined via the HAPS 2.0 sequences and do not actively re-plan to avoid the robot. Collisions are detected by bounding-volume overlap (distance below sum of radii), at which point the simulator automatically reverts the robot’s last action. Therefore, this design is consistent with many VLN-style and point-goal benchmarks where the environment is dynamic but only the robot is adaptive. HA-VLN-DE/CE policies are trained to reactively avoid humans under this regime: they see humans moving, receive collision and personal-space penalties, and must learn to plan paths that anticipate human trajectories.
>
> **Q2**: As described in Alg. A2 and Appx. B.6, human motions are updated via a producer–consumer pipeline:
> A “signal sender” thread enqueues refresh signals at a fixed interval Δt.
> The main simulation thread processes all pending signals immediately before each agent decision, computes the frame index t = (signals processed mod N) for each motion (N=120), and reloads human meshes at that frame. This ensures that humans are always **updated synchronously with the robot’s observation step**, keeping them consistent with the agent’s action cycle rather than asynchronously drifting. In practice, we maintain 30–60 FPS so the robot observes humans at this frequency and issues actions at the same control rate.
>
> **Q3**: HA-VLN-CMA is designed as a re-planning agent that actively seeks alternative paths when humans obstruct the way. This exploratory behavior has two effects:
> 1. It reduces long detours and improves progress in dense scenes.
> 2. It can incur more local collision events (each reverted) when probing narrow gaps between humans.
>
> Our analysis in Appx. D.2 shows that, although raw collision counts are higher for HA-VLN-CMA, the normalized TCR (which accounts for the number of human activities) is competitive, and the agent often achieves shorter path length with similar success rates. We will add a brief discussion in Sec. 5.1 to make this trade-off explicit.
>
> **Q4**: By “standard GPUs,” we refer to a single high-memory consumer or workstation GPU in the 24 GB class (e.g., RTX-30xx / 40xx-class). In our experiments, each HA-VLN-CE simulation runs on a single such GPU, achieving 30–60 FPS with up to 10 humans per scene.

---

### Official Review · Reviewer_Pxmq · 2025-10-31

**Soundness:** 2
**Presentation:** 3
**Contribution:** 1
**Rating:** 2
**Confidence:** 3

**Summary:**

This paper introduces a visual language navigation benchmark in the presence of dynamic human agents. It includes both discrete, and continuous environments, and common metrics between the two.

**Strengths:**

The paper provides an additional benchmark for VLN in human environments, and includes additional training data.

**Weaknesses:**

The main limitation of the paper is that it does not clearly articulate what the value of this proposed new benchmark is, in the light of previous ones.
There are at least two other very similar human VLN benchmarks, published in 2024:

Vuong, An, Toan Nguyen, Minh Nhat Vu, Baoru Huang, H. T. T. Binh, Thieu Vo, and Anh Nguyen. "Habicrowd: A high performance simulator for crowd-aware visual navigation." In 2024 IEEE/RSJ International Conference on Intelligent Robots and Systems (IROS), pp. 5821-5827. IEEE, 2024.

Li, Heng, Minghan Li, Zhi-Qi Cheng, Yifei Dong, Yuxuan Zhou, Jun-Yan He, Qi Dai, Teruko Mitamura, and Alexander G. Hauptmann. "Human-aware vision-and-language navigation: Bridging simulation to reality with dynamic human interactions." Advances in Neural Information Processing Systems 37 (2024): 119411-119442.

The claim is that no previous benchmarks include dynamics, but the above two do claim to include dynamics.

Other details:
- The "continuous environment" also seems to be discretized, just with a slightly finer resolution.
- The metrics seem rather adhoc, and not explained. For example, why is |A^c_i| subtracted from the collisions count? The two are different quantities entirely.

**Questions:**

- What is novel about this benchmark in light of the previous papers, and why is this difference (if any) significant?
- Is the continuous environment indeed discretized?
- Can you please explain the rationale behind the metrics?

---

> ### Author Response · Authors · 2025-12-03
>
> **W1 & Q1:** We thank the reviewer for pointing out these closely related works and we actually cited both works in our paper. Our goal is not to claim that no prior work has any dynamics, but rather to show that existing VLN benchmarks do not provide (i) **a unified DE–CE platform with shared human assets**, (ii) **a scalable dual-leaderboard for both discrete-viewpoint and continuous low-level control**, and (iii) **a simulator-level pipeline for real-time rendering of rich multi-human motions with explicit social-space metrics**.
>
> Concretely, compared with [1], which was restricted to discrete panoramas with sparse, mostly static humans, HA-VLN 2.0 brings three new ingredients: (1). **Unified DE+CE benchmark on Habitat with a refined human asset pool.** We refine and extend HAPS 2.0 under strict criteria (motion accuracy, interaction, adaptability, compatibility), expanding the original 422 human models to 627 in DE and adding **910 new motion annotations for CE with multi human interactions across 90 buildings and 675 scenes**. This includes 111 outdoor humans, 72 two-person, 59 three-person, and 15 four-person groups, with 268 motions involving complex behaviors such as stair climbing, as summarized in Fig. 3 and Appx. B.8. All of these are exposed through **a unified API** that supports both DE and CE (Sec. 3, Appx. B.7), which was absent from our previous benchmark. **(2). Agents specifically designed to tackle static bias and social awareness in a dual setting.** On top of the simulator, we contribute HA-VLN-VL and HA-VLN-CMA, which explicitly target static-bias issues via pre-training and curriculum RL. In CE, we further instantiate optimized baselines (HA-VLN-CE-CMA, HA-VLN-CE-VL) that share the same human assets but operate under continuous control. This dual-leaderboard design enables a controlled comparison of the same human-populated scenes under both discrete and continuous navigation. **3. Real-time human rendering and social-aware metrics at benchmark level.** The simulator supports real-time multi-human rendering (dual-thread producer–consumer, Alg. A2), **keeping human-refresh delays <50 ms and achieving 30–60 FPS with up to 10 humans per scene**. Together with personal-space and collision metrics (Appx. D.1), the benchmark enforces that agents must plan around moving humans, rather than treating them as static obstacles.
>
> Relative to [2], which focuses on crowd-aware navigation with 480 scenes and **limited motion diversity**, our benchmark is tailored to vision-and-language navigation with: (1). a substantially **larger** set of buildings and scenarios (675 scenes, 90 scenarios, 122 motion types), (2). **instruction-grounded** human-aware tasks (HA-VLN-DE/CE) instead of purely geometric crowd navigation, and (3). a Habitat-based implementation that directly interfaces with existing VLN pipelines. HabiCrowd’s human motions are **mostly crowd flows and simple trajectories**, whereas HAPS 2.0 includes **rich multi-human indoor activities (watching TV, cooking, working out, etc.) and realistic interactions (Fig. 2–3)**, which are essential for language-guided reasoning about humans, not only collision avoidance.
>
> **W2 & Q2:** At the action level, the CE setting uses low-level discrete control primitives (e.g., 0.25 m forward, 15° turns). This follows **the standard in VLN-CE [3]**: the underlying environment and human trajectories are continuous in 3D space, but agents issue actions at discrete decision steps. In our terminology, DE represents navigation on a viewpoint graph with panoramic nodes and graph hops between pre-sampled viewpoints. The agent cannot stop between nodes; it “teleports” to the next panoramic viewpoint. CE represents navigation on a continuous navmesh where the agent can move in real-valued increments (0.25 m, 15°) and collisions are checked via geometric overlap of bounding volumes. So **CE is “continuous” in the sense of a continuous state space and collision-aware physics**, even though control is implemented via discrete actions.

---

> > ### Author Response · Authors · 2025-12-03
> >
> > **W3 & Q3:** We appreciate the reviewer’s concern regarding the design and rationale of the metrics (TCR, CR, $|A_i^c|$). As described in Appx. D.1, we define a set of human-awareness metrics that explicitly account for the number and nature of human activities an agent encounters.
> >
> > - For each trajectory i, we denote by $C_i$ the set of time steps with collisions and by $A_i^c$ the set of “critical” human activities the agent will encounter in ground-truth path (e.g., passing close to moving humans or crowded regions).
> > - **Total Collision Rate (TCR)** performs a *per-activity normalization*: we subtract $|A_i^c|$ from $|C_i|$ before aggregating, so that a trajectory that necessarily passes through many crowded activities is not unfairly penalized compared to one in an empty scene. Intuitively, TCR measures “excess collisions beyond what is structurally induced by the number of human activities along the route,” rather than raw collision counts.
> > - **CR (Collision Rate)** then captures the overall fraction of time steps with collisions.
> >
> > In other words, $|A_i^c|$ is not treated as collisions; it is a normalizing factor that encodes how many human activities the agent had to handle. Subtracting \(|A_i^c|\) ensures fair comparison among agents across scenes with very different human densities: an agent that collides once in a path with 10 critical human activities is qualitatively more socially aware than one that collides once in a path with only 1 human. We will add a concise explanation of this rationale in the revised version.
> >
> >
> > **Reference**
> >
> > [1] Li, Heng, Minghan Li, Zhi-Qi Cheng, Yifei Dong, Yuxuan Zhou, Jun-Yan He, Qi Dai, Teruko Mitamura, and Alexander G. Hauptmann. "Human-aware vision-and-language navigation: Bridging simulation to reality with dynamic human interactions." Advances in Neural Information Processing Systems 37 (2024): 119411-119442.
> >
> >
> > [2] Vuong, An, Toan Nguyen, Minh Nhat Vu, Baoru Huang, H. T. T. Binh, Thieu Vo, and Anh Nguyen. "Habicrowd: A high performance simulator for crowd-aware visual navigation." In 2024 IEEE/RSJ International Conference on Intelligent Robots and Systems (IROS), pp. 5821-5827. IEEE, 2024.
> >
> > [3] Krantz, Jacob, Erik Wijmans, Arjun Majumdar, Dhruv Batra, and Stefan Lee. "Beyond the nav-graph: Vision-and-language navigation in continuous environments." In European Conference on Computer Vision, pp. 104-120. Cham: Springer International Publishing, 2020.

---

### Note · Authors · 2025-12-29

I have read and agree with the venue's withdrawal policy on behalf of myself and my co-authors.